# Data Programming:
# Creating Large Training Sets, Quickly

**Alexander Ratner, Christopher De Sa, Sen Wu, Daniel Selsam, Christopher Ré**
Stanford University
{ajratner,cdesa,senwu,dselsam,chrismre}@stanford.edu

## Abstract

Large labeled training sets are the critical building blocks of supervised learning methods and are key enablers of deep learning techniques. For some applications, creating labeled training sets is the most time-consuming and expensive part of applying machine learning. We therefore propose a paradigm for the programmatic creation of training sets called *data programming* in which users express weak supervision strategies or domain heuristics as *labeling functions*, which are programs that label subsets of the data, but that are noisy and may conflict. We show that by explicitly representing this training set labeling process as a generative model, we can "denoise" the generated training set, and establish theoretically that we can recover the parameters of these generative models in a handful of settings. We then show how to modify a discriminative loss function to make it noise-aware, and demonstrate our method over a range of discriminative models including logistic regression and LSTMs. Experimentally, on the 2014 TAC-KBP Slot Filling challenge, we show that data programming would have led to a new winning score, and also show that applying data programming to an LSTM model leads to a TAC-KBP score almost 6 F1 points over a state-of-the-art LSTM baseline (and into second place in the competition). Additionally, in initial user studies we observed that data programming may be an easier way for non-experts to create machine learning models when training data is limited or unavailable.

## 1 Introduction

Many of the major machine learning breakthroughs of the last decade have been catalyzed by the release of a new labeled training dataset.[1] Supervised learning approaches that use such datasets have increasingly become key building blocks of applications throughout science and industry. This trend has also been fueled by the recent empirical success of automated feature generation approaches, notably deep learning methods such as long short term memory (LSTM) networks [14], which ameliorate the burden of feature engineering given large enough labeled training sets. For many real-world applications, however, large hand-labeled training sets do not exist, and are prohibitively expensive to create due to requirements that labelers be experts in the application domain. Furthermore, applications' needs often change, necessitating new or modified training sets.

To help reduce the cost of training set creation, we propose *data programming*, a paradigm for the programmatic creation and modeling of training datasets. Data programming provides a simple, unifying framework for *weak supervision*, in which training labels are noisy and may be from multiple, potentially overlapping sources. In data programming, users encode this weak supervision in the form of *labeling functions*, which are user-defined programs that each provide a label for some subset of the data, and collectively generate a large but potentially overlapping set of training labels. Many different weak supervision approaches can be expressed as labeling functions, such

as strategies which utilize existing knowledge bases (as in distant supervision [22]), model many individual annotator's labels (as in crowdsourcing), or leverage a combination of domain-specific patterns and dictionaries. Because of this, labeling functions may have widely varying error rates and may conflict on certain data points. To address this, we model the labeling functions as a generative process, which lets us automatically denoise the resulting training set by learning the accuracies of the labeling functions along with their correlation structure. In turn, we use this model of the training set to optimize a stochastic version of the loss function of the discriminative model that we desire to train. We show that, given certain conditions on the labeling functions, our method achieves the same asymptotic scaling as supervised learning methods, but that our scaling depends on the amount of *unlabeled* data, and uses only a fixed number of labeling functions.

Data programming is in part motivated by the challenges that users faced when applying prior programmatic supervision approaches, and is intended to be a new software engineering paradigm for the creation and management of training sets. For example, consider the scenario when two labeling functions of differing quality and scope overlap and possibly conflict on certain training examples; in prior approaches the user would have to decide which one to use, or how to somehow integrate the signal from both. In data programming, we accomplish this automatically by learning a model of the training set that includes both labeling functions. Additionally, users are often aware of, or able to induce, dependencies between their labeling functions. In data programming, users can provide a dependency graph to indicate, for example, that two labeling functions are similar, or that one "fixes" or "reinforces" another. We describe cases in which we can learn the strength of these dependencies, and for which our generalization is again asymptotically identical to the supervised case.

One further motivation for our method is driven by the observation that users often struggle with selecting *features* for their models, which is a traditional development bottleneck given fixed-size training sets. However, initial feedback from users suggests that writing labeling functions in the framework of data programming may be easier [12]. While the impact of a feature on end performance is dependent on the training set and on statistical characteristics of the model, a labeling function has a simple and intuitive optimality criterion: that it labels data correctly. Motivated by this, we explore whether we can flip the traditional machine learning development process on its head, having users instead focus on generating training sets large enough to support automatically-generated features.

**Summary of Contributions and Outline** Our first contribution is the *data programming* framework, in which users can implicitly describe a rich generative model for a training set in a more flexible and general way than in previous approaches. In Section 3, we first explore a simple model in which labeling functions are conditionally independent. We show here that under certain conditions, the sample complexity is nearly the same as in the labeled case. In Section 4, we extend our results to more sophisticated data programming models, generalizing related results in crowdsourcing [17]. In Section 5, we validate our approach experimentally on large real-world text relation extraction tasks in genomics, pharmacogenomics and news domains, where we show an average 2.34 point F1 score improvement over a baseline distant supervision approach—including what would have been a new competition-winning score for the 2014 TAC-KBP Slot Filling competition. Using LSTM-generated features, we additionally would have placed second in this competition, achieving a 5.98 point F1 score gain over a state-of-the-art LSTM baseline [32]. Additionally, we describe promising feedback from a usability study with a group of bioinformatics users.

## 2   Related Work

Our work builds on many previous approaches in machine learning. *Distant supervision* is one approach for programmatically creating training sets. The canonical example is relation extraction from text, wherein a knowledge base of known relations is heuristically mapped to an input corpus [8, 22]. Basic extensions group examples by surrounding textual patterns, and cast the problem as a *multiple instance learning* one [15, 25]. Other extensions model the accuracy of these surrounding textual patterns using a discriminative feature-based model [26], or generative models such as hierarchical topic models [1, 27, 31]. Like our approach, these latter methods model a generative process of training set creation, however in a proscribed way that is not based on user input as in our approach. There is also a wealth of examples where additional heuristic patterns used to label training data are collected from unlabeled data [7] or directly from users [21, 29], in a similar manner to our approach, but without any framework to deal with the fact that said labels are explicitly noisy.

*Crowdsourcing* is widely used for various machine learning tasks [13, 18]. Of particular relevance to our problem setting is the theoretical question of how to model the accuracy of various experts without ground truth available, classically raised in the context of crowdsourcing [10]. More recent results provide formal guarantees even in the absence of labeled data using various approaches [4, 9, 16, 17, 24, 33]. Our model can capture the basic model of the crowdsourcing setting, and can be considered equivalent in the independent case (Sec. 3). However, in addition to generalizing beyond getting inputs solely from human annotators, we also model user-supplied dependencies between the "labelers" in our model, which is not natural within the context of crowdsourcing. Additionally, while crowdsourcing results focus on the regime of a large number of labelers each labeling a small subset of the data, we consider a small set of labeling functions each labeling a large portion of the dataset.

*Co-training* is a classic procedure for effectively utilizing both a small amount of labeled data and a large amount of unlabeled data by selecting two conditionally independent *views* of the data [5]. In addition to not needing a set of labeled data, and allowing for more than two views (labeling functions in our case), our approach allows explicit modeling of dependencies between views, for example allowing observed issues with dependencies between views to be explicitly modeled [19].

*Boosting* is a well known procedure for combining the output of many "weak" classifiers to create a strong classifier in a supervised setting [28]. Recently, boosting-like methods have been proposed which leverage unlabeled data in addition to labeled data, which is also used to set constraints on the accuracies of the individual classifiers being ensembled [3]. This is similar in spirit to our approach, except that labeled data is not explicitly necessary in ours, and richer dependency structures between our "heuristic" classifiers (labeling functions) are supported.

The general case of *learning with noisy labels* is treated both in classical [20] and more recent contexts [23]. It has also been studied specifically in the context of *label-noise robust* logistic regression [6]. We consider the more general scenario where multiple noisy labeling functions can conflict and have dependencies.

## 3  The Data Programming Paradigm

In many applications, we would like to use machine learning, but we face the following challenges: (i) *hand-labeled* training data is not available, and is prohibitively expensive to obtain in sufficient quantities as it requires expensive domain expert labelers; (ii) *related external knowledge bases* are either unavailable or insufficiently specific, precluding a traditional distant supervision or co-training approach; (iii) *application specifications* are in flux, changing the model we ultimately wish to learn.

In such a setting, we would like a simple, scalable and adaptable approach for supervising a model applicable to our problem. More specifically, we would ideally like our approach to achieve $\epsilon$ expected loss with high probability, given $O(1)$ *inputs* of some sort from a domain-expert user, rather than the traditional $\tilde{O}(\epsilon^{-2})$ hand-labeled training examples required by most supervised methods (where $\tilde{O}$ notation hides logarithmic factors). To this end, we propose *data programming*, a paradigm for the programmatic creation of training sets, which enables domain-experts to more rapidly train machine learning systems and has the potential for this type of scaling of expected loss. In data programming, rather than manually labeling each example, users instead describe the *processes by which* these points could be labeled by providing a set of heuristic rules called *labeling functions*.

In the remainder of this paper, we focus on a binary classification task in which we have a distribution $\pi$ over object and class pairs $(x, y) \in X \times \{-1, 1\}$, and we are concerned with minimizing the logistic loss under a linear model given some *features*,

$$l(w) = \mathbf{E}_{(x,y) \sim \pi} \left[ \log(1 + \exp(-w^T f(x)y)) \right],$$

where without loss of generality, we assume that $\|f(x)\| \le 1$. Then, a labeling function $\lambda_i : X \mapsto \{-1, 0, 1\}$ is a user-defined function that encodes some domain heuristic, which provides a (non-zero) label for some subset of the objects. As part of a *data programming specification*, a user provides some $m$ labeling functions, which we denote in vectorized form as $\lambda : X \mapsto \{-1, 0, 1\}^m$.

**Example 3.1.** To gain intuition about labeling functions, we describe a simple text relation extraction example. In Figure 1, we consider the task of classifying co-occurring gene and disease mentions as either expressing a causal relation or not. For example, given the sentence "Gene A causes disease B", the object $x = (A, B)$ has true class $y = 1$. To construct a training set, the user writes three labeling

```
def lambda_1(x):
    return 1 if (x.gene,x.pheno) in KNOWN_RELATIONS_1 else 0

def lambda_2(x):
    return -1 if re.match(r'.*not_cause.*', x.text_between) else 0

def lambda_3(x):
    return 1 if re.match(r'.*associated.*', x.text_between)
          and (x.gene,x.pheno) in KNOWN_RELATIONS_2 else 0
```

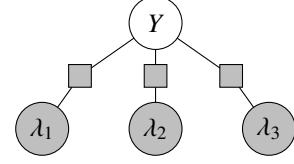

(b) The generative model of a training set defined by the user input (unary factors omitted).

(a) An example set of three labeling functions written by a user.

Figure 1: An example of extracting mentions of gene-disease relations from the scientific literature.

functions (Figure 1a). In $\lambda_1$, an external structured knowledge base is used to label a few objects with relatively high accuracy, and is equivalent to a traditional distant supervision rule (see Sec. 2). $\lambda_2$ uses a purely heuristic approach to label a much larger number of examples with lower accuracy. Finally, $\lambda_3$ is a "hybrid" labeling function, which leverages a knowledge base and a heuristic.

A labeling function need not have perfect accuracy or recall; rather, it represents a pattern that the user wishes to impart to their model and that is easier to encode as a labeling function than as a set of hand-labeled examples. As illustrated in Ex. 3.1, labeling functions can be based on external knowledge bases, libraries or ontologies, can express heuristic patterns, or some hybrid of these types; we see evidence for the existence of such diversity in our experiments (Section 5). The use of labeling functions is also strictly more general than manual annotations, as a manual annotation can always be directly encoded by a labeling function. Importantly, labeling functions can overlap, conflict, and even have dependencies which users can provide as part of the data programming specification (see Section 4); our approach provides a simple framework for these inputs.

**Independent Labeling Functions**   We first describe a model in which the labeling functions label independently, given the true label class. Under this model, each labeling function $\lambda_i$ has some probability $\beta_i$ of labeling an object and then some probability $\alpha_i$ of labeling the object correctly; for simplicity we also assume here that each class has probability 0.5. This model has distribution

$$\mu_{\alpha,\beta}(\Lambda, Y) = \frac{1}{2} \prod_{i=1}^{m} \left( \beta_i \alpha_i \mathbf{1}_{\{\Lambda_i=Y\}} + \beta_i(1 - \alpha_i)\mathbf{1}_{\{\Lambda_i=-Y\}} + (1 - \beta_i)\mathbf{1}_{\{\Lambda_i=0\}} \right), \qquad (1)$$

where $\Lambda \in \{-1, 0, 1\}^m$ contains the labels output by the labeling functions, and $Y \in \{-1, 1\}$ is the predicted class. If we allow the parameters $\alpha \in \mathbb{R}^m$ and $\beta \in \mathbb{R}^m$ to vary, (1) specifies a family of generative models. In order to expose the scaling of the expected loss as the size of the unlabeled dataset changes, we will assume here that $0.3 \leq \beta_i \leq 0.5$ and $0.8 \leq \alpha_i \leq 0.9$. We note that while these arbitrary constraints can be changed, they are roughly consistent with our applied experience, where users tend to write high-accuracy and high-coverage labeling functions.

Our first goal will be to learn which parameters $(\alpha, \beta)$ are most consistent with our observations—our unlabeled training set—using maximum likelihood estimation. To do this for a particular training set $S \subset \mathcal{X}$, we will solve the problem

$$(\hat{\alpha}, \hat{\beta}) = \arg\max_{\alpha,\beta} \sum_{x \in S} \log \mathbf{P}_{(\Lambda,Y)\sim\mu_{\alpha,\beta}} (\Lambda = \lambda(x)) = \arg\max_{\alpha,\beta} \sum_{x \in S} \log \left( \sum_{y' \in \{-1,1\}} \mu_{\alpha,\beta}(\lambda(x), y') \right) \qquad (2)$$

In other words, we are maximizing the probability that the observed labels produced on our training examples occur under the generative model in (1). In our experiments, we use stochastic gradient descent to solve this problem; since this is a standard technique, we defer its analysis to the appendix.

**Noise-Aware Empirical Loss**   Given that our parameter learning phase has successfully found some $\hat{\alpha}$ and $\hat{\beta}$ that accurately describe the training set, we can now proceed to estimate the parameter $w$ which minimizes the expected risk of a linear model over our feature mapping $f$, given $\hat{\alpha}, \hat{\beta}$. To do so, we define the *noise-aware empirical risk* $L_{\hat{\alpha},\hat{\beta}}$ with regularization parameter $\rho$, and compute the *noise-aware empirical risk minimizer*

$$\hat{w} = \arg\min_{w} L_{\hat{\alpha},\hat{\beta}}(w; S) = \arg\min_{w} \frac{1}{|S|} \sum_{x \in S} \mathbf{E}_{(\Lambda,Y)\sim\mu_{\hat{\alpha},\hat{\beta}}} \left[ \log \left( 1 + e^{-w^T f(x)Y} \right) \Big| \Lambda = \lambda(x) \right] + \rho \|w\|^2 \qquad (3)$$

This is a logistic regression problem, so it can be solved using stochastic gradient descent as well.

We can in fact prove that stochastic gradient descent running on (2) and (3) is guaranteed to produce accurate estimates, under conditions which we describe now. First, the problem distribution $\pi$ needs to be accurately modeled by some distribution $\mu$ in the family that we are trying to learn. That is, for some $\alpha^*$ and $\beta^*$,

$$\forall \Lambda \in \{-1, 0, 1\}^m, Y \in \{-1, 1\}, \mathbf{P}_{(x,y)\sim\pi^*}(\lambda(x) = \Lambda, y = Y) = \mu_{\alpha^*,\beta^*}(\Lambda, Y). \tag{4}$$

Second, given an example $(x, y) \sim \pi^*$, the class label $y$ must be independent of the features $f(x)$ given the labels $\lambda(x)$. That is,

$$(x, y) \sim \pi^* \Rightarrow y \perp f(x) \mid \lambda(x). \tag{5}$$

This assumption encodes the idea that the labeling functions, while they may be arbitrarily dependent on the features, provide sufficient information to accurately identify the class. Third, we assume that the algorithm used to solve (3) has bounded generalization risk such that for some parameter $\chi$,

$$\mathbf{E}_{\hat{w}}\left[\mathbf{E}_S\left[L_{\hat{\alpha},\hat{\beta}}(\hat{w}; S)\right] - \min_w \mathbf{E}_S\left[L_{\hat{\alpha},\hat{\beta}}(w; S)\right]\right] \leq \chi. \tag{6}$$

Under these conditions, we make the following statement about the accuracy of our estimates, which is a simplified version of a theorem that is detailed in the appendix.

**Theorem 1.** *Suppose that we run data programming, solving the problems in (2) and (3) using stochastic gradient descent to produce $(\hat{\alpha}, \hat{\beta})$ and $\hat{w}$. Suppose further that our setup satisfies the conditions (4), (5), and (6), and suppose that $m \geq 2000$. Then for any $\epsilon > 0$, if the number of labeling functions $m$ and the size of the input dataset $S$ are large enough that*

$$|S| \geq \frac{356}{\epsilon^2} \log\left(\frac{m}{3\epsilon}\right)$$

*then our expected parameter error and generalization risk can be bounded by*

$$\mathbf{E}\left[\|\hat{\alpha} - \alpha^*\|^2\right] \leq m\epsilon^2 \qquad \mathbf{E}\left[\|\hat{\beta} - \beta^*\|^2\right] \leq m\epsilon^2 \qquad \mathbf{E}\left[l(\hat{w}) - \min_w l(w)\right] \leq \chi + \frac{\epsilon}{27\rho}.$$

We select $m \geq 2000$ to simplify the statement of the theorem and give the reader a feel for how $\epsilon$ scales with respect to $|S|$. The full theorem with scaling in each parameter (and for arbitrary $m$) is presented in the appendix. This result establishes that to achieve both expected loss and parameter estimate error $\epsilon$, it suffices to have only $m = O(1)$ labeling functions and $|S| = \tilde{O}(\epsilon^{-2})$ training examples, which is the same asymptotic scaling exhibited by methods that use labeled data. This means that data programming achieves the same learning rate as methods that use labeled data, while requiring asymptotically less work from its users, who need to specify $O(1)$ labeling functions rather than manually label $\tilde{O}(\epsilon^{-2})$ examples. In contrast, in the crowdsourcing setting [17], the number of workers $m$ tends to infinity while here it is constant while the dataset grows. These results provide some explanation of why our experimental results suggest that a small number of rules with a large unlabeled training set can be effective at even complex natural language processing tasks.

## 4 Handling Dependencies

In our experience with data programming, we have found that users often write labeling functions that have clear dependencies among them. As more labeling functions are added as the system is developed, an implicit dependency structure arises naturally amongst the labeling functions: modeling these dependencies can in some cases improve accuracy. We describe a method by which the user can specify this dependency knowledge as a *dependency graph*, and show how the system can use it to produce better parameter estimates.

**Label Function Dependency Graph** To support the injection of dependency information into the model, we augment the data programming specification with a *label function dependency graph*, $G \subset \mathcal{D} \times \{1, \ldots, m\} \times \{1, \ldots, m\}$, which is a directed graph over the labeling functions, each of the edges of which is associated with a *dependency type* from a class of dependencies $\mathcal{D}$ appropriate to the domain. From our experience with practitioners, we identified four commonly-occurring types of dependencies as illustrative examples: *similar*, *fixing*, *reinforcing*, and *exclusive* (see Figure 2).

For example, suppose that we have two functions $\lambda_1$ and $\lambda_2$, and $\lambda_2$ typically labels only when (i) $\lambda_1$ also labels, (ii) $\lambda_1$ and $\lambda_2$ disagree in their labeling, and (iii) $\lambda_2$ is actually correct. We call this a fixing dependency, since $\lambda_2$ *fixes* mistakes made by $\lambda_1$. If $\lambda_1$ and $\lambda_2$ were to typically agree rather than disagree, this would be a reinforcing dependency, since $\lambda_2$ reinforces a subset of the labels of $\lambda_1$.

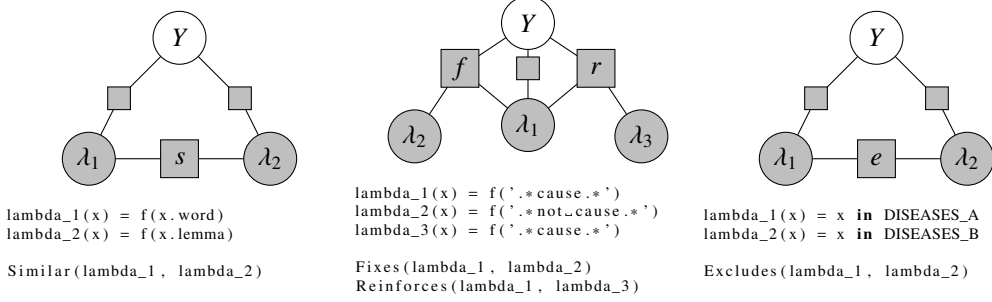

Figure 2: Examples of labeling function dependency predicates.

**Modeling Dependencies**   The presence of dependency information means that we can no longer model our labels using the simple Bayesian network in (1). Instead, we model our distribution as a factor graph. This standard technique lets us describe the family of generative distributions in terms of a known *factor function* $h : \{-1, 0, 1\}^m \times \{-1, 1\} \mapsto \{-1, 0, 1\}^M$ (in which each entry $h_i$ represents a factor), and an unknown parameter $\theta \in \mathbb{R}^M$ as

$$\mu_\theta(\Lambda, Y) = Z_\theta^{-1} \exp(\theta^T h(\Lambda, Y)),$$

where $Z_\theta$ is the *partition function* which ensures that $\mu$ is a distribution. Next, we will describe how we define $h$ using information from the dependency graph.

To construct $h$, we will start with some base factors, which we inherit from (1), and then augment them with additional factors representing dependencies. For all $i \in \{1, \dots, m\}$, we let

$$h_0(\Lambda, Y) = Y, \quad h_i(\Lambda, Y) = \Lambda_i Y, \quad h_{m+i}(\Lambda, Y) = \Lambda_i, \quad h_{2m+i}(\Lambda, Y) = \Lambda_i^2 Y, \quad h_{3m+i}(\Lambda, Y) = \Lambda_i^2.$$

These factors alone are sufficient to describe any distribution for which the labels are mutually independent, given the class: this includes the independent family in (1).

We now proceed by adding additional factors to $h$, which model the dependencies encoded in $G$. For each dependency edge $(d, i, j)$, we add one or more factors to $h$ as follows. For a near-duplicate dependency on $(i, j)$, we add a single factor $h_t(\Lambda, Y) = \mathbf{1}\{\Lambda_i = \Lambda_j\}$, which increases our prior probability that the labels will agree. For a fixing dependency, we add two factors, $h_t(\Lambda, Y) = -\mathbf{1}\{\Lambda_i = 0 \wedge \Lambda_j \neq 0\}$ and $h_{t+1}(\Lambda, Y) = \mathbf{1}\{\Lambda_i = -Y \wedge \Lambda_j = Y\}$, which encode the idea that $\lambda_j$ labels only when $\lambda_i$ does, and that $\lambda_j$ fixes errors made by $\lambda_i$. The factors for a reinforcing dependency are the same, except that $h_{t+1}(\Lambda, Y) = \mathbf{1}\{\Lambda_i = Y \wedge \Lambda_j = Y\}$. Finally, for an exclusive dependency, we have a single factor $h_t(\Lambda, Y) = -\mathbf{1}\{\Lambda_i \neq 0 \wedge \Lambda_j \neq 0\}$.

**Learning with Dependencies**   We can again solve a maximum likelihood problem like (2) to learn the parameter $\hat\theta$. Using the results, we can continue on to find the noise-aware empirical loss minimizer by solving the problem in (3). In order to solve these problems in the dependent case, we typically invoke stochastic gradient descent, using Gibbs sampling to sample from the distributions used in the gradient update. Under conditions similar to those in Section 3, we can again provide a bound on the accuracy of these results. We define these conditions now. First, there must be some set $\Theta \subset \mathbb{R}^M$ that we know our parameter lies in. This is analogous to the assumptions on $\alpha_i$ and $\beta_i$ we made in Section 3, and we can state the following analogue of (4):

$$\exists \theta^* \in \Theta \text{ s.t. } \forall (\Lambda, Y) \in \{-1, 0, 1\}^m \times \{-1, 1\}, \mathbf{P}_{(x,y)\sim\pi^*}(\lambda(x) = \Lambda, y = Y) = \mu_{\theta^*}(\Lambda, Y). \tag{7}$$

Second, for any $\theta \in \Theta$, it must be possible to accurately learn $\theta$ from full (i.e. labeled) samples of $\mu_\theta$. More specifically, there exists an unbiased estimator $\hat\theta(T)$ that is a function of some dataset $T$ of independent samples from $\mu_\theta$ such that, for some $c > 0$ and for all $\theta \in \Theta$,

$$\mathbf{Cov}\left(\hat\theta(T)\right) \preceq (2c |T|)^{-1} I. \tag{8}$$

Third, for any two feasible models $\theta_1$ and $\theta_2 \in \Theta$,

$$\mathbf{E}_{(\Lambda_1, Y_1)\sim\mu_{\theta_1}}\left[\mathbf{Var}_{(\Lambda_2, Y_2)\sim\mu_{\theta_2}}(Y_2|\Lambda_1 = \Lambda_2)\right] \leq c M^{-1}. \tag{9}$$

That is, we'll usually be reasonably sure in our guess for the value of $Y$, even if we guess using distribution $\mu_{\theta_2}$ while the the labeling functions were actually sampled from (the possibly totally different) $\mu_{\theta_1}$. We can now prove the following result about the accuracy of our estimates.

| Features | Method | KBP (News) | | | Genomics | | | Pharmacogenomics | | |
|---|---|---|---|---|---|---|---|---|---|---|
| | | Prec. | Rec. | F1 | Prec. | Rec. | F1 | Prec. | Rec. | F1 |
| Hand-tuned | ITR | **51.15** | 26.72 | 35.10 | 83.76 | 41.67 | 55.65 | 68.16 | 49.32 | 57.23 |
| | DP | 50.52 | **29.21** | **37.02** | **83.90** | 43.43 | 57.24 | **68.36** | 54.80 | **60.83** |
| LSTM | ITR | 37.68 | 28.81 | 32.66 | 69.07 | **50.76** | 58.52 | 32.35 | 43.84 | 37.23 |
| | DP | 47.47 | 27.88 | 35.78 | 75.48 | 48.48 | **58.99** | 37.63 | 47.95 | 42.17 |

Table 1: Precision/Recall/F1 scores using data programming (DP), as compared to distant supervision ITR approach, with both hand-tuned and LSTM-generated features.

**Theorem 2.** *Suppose that we run stochastic gradient descent to produce $\hat{\theta}$ and $\hat{w}$, and that our setup satisfies the conditions (5)-(9). Then for any $\epsilon > 0$, if the input dataset $S$ is large enough that*

$$|S| \geq \frac{2}{c^2 \epsilon^2} \log\left(\frac{2 \|\theta_0 - \theta^*\|^2}{\epsilon}\right),$$

*then our expected parameter error and generalization risk can be bounded by*

$$\mathbf{E}\left[\left\|\hat{\theta} - \theta^*\right\|^2\right] \leq M\epsilon^2 \qquad\qquad \mathbf{E}\left[l(\hat{w}) - \min_w l(w)\right] \leq \chi + \frac{c\epsilon}{2\rho}.$$

As in the independent case, this shows that we need only $|S| = \tilde{O}(\epsilon^{-2})$ unlabeled training examples to achieve error $O(\epsilon)$, which is the same asymptotic scaling as supervised learning methods. This suggests that while we pay a computational penalty for richer dependency structures, we are no less statistically efficient. In the appendix, we provide more details, including an explicit description of the algorithm and the step size used to achieve this result.

## 5 Experiments

We seek to experimentally validate three claims about our approach. Our first claim is that data programming can be an effective paradigm for building high quality machine learning systems, which we test across three real-world relation extraction applications. Our second claim is that data programming can be used successfully in conjunction with automatic feature generation methods, such as LSTM models. Finally, our third claim is that data programming is an intuitive and productive framework for domain-expert users, and we report on our initial user studies.

**Relation Mention Extraction Tasks** In the *relation mention extraction* task, our objects are relation mention *candidates* $x = (e_1, e_2)$, which are pairs of entity mentions $e_1, e_2$ in unstructured text, and our goal is to learn a model that classifies each candidate as either a true textual assertion of the relation $R(e_1, e_2)$ or not. We examine a news application from the 2014 TAC-KBP Slot Filling challenge[2], where we extract relations between real-world entities from articles [2]; a clinical genomics application, where we extract causal relations between genetic mutations and phenotypes from the scientific literature[3]; and a pharmacogenomics application where we extract interactions between genes, also from the scientific literature [21]; further details are included in the Appendix.

For each application, we or our collaborators originally built a system where a training set was programmatically generated by ordering the labeling functions as a sequence of if-then-return statements, and for each candidate, taking the first label emitted by this script as the training label. We refer to this as the *if-then-return (ITR)* approach, and note that it often required significant domain expert development time to tune (weeks or more). For this set of experiments, we then used the same labeling function sets within the framework of data programming. For all experiments, we evaluated on a blind hand-labeled evaluation set. In Table 1, we see that we achieve consistent improvements: on average by 2.34 points in F1 score, including what would have been a winning score on the 2014 TAC-KBP challenge [30].

We observed these performance gains across applications with very different labeling function sets. We describe the labeling function summary statistics—*coverage* is the percentage of objects that had at least one label, *overlap* is the percentage of objects with more than one label, and *conflict* is

the percentage of objects with conflicting labels—and see in Table 2 that even in scenarios where *m* is small, and conflict and overlap is relatively less common, we still realize performance gains. Additionally, on a disease mention extraction task (see Usability Study), which was written from scratch within the data programming paradigm, allowing developers to supply dependencies of the basic types outlined in Sec. 4 led to a 2.3 point F1 score boost.

| Application | # of LFs | Coverage | $|S_{\lambda \neq 0}|$ | Overlap | Conflict | F1 Score Improvement | |
|---|---|---|---|---|---|---|---|
| | | | | | | HT | LSTM |
| KBP (News) | 40 | 29.39 | 2.03M | 1.38 | 0.15 | 1.92 | 3.12 |
| Genomics | 146 | 53.61 | 256K | 26.71 | 2.05 | 1.59 | 0.47 |
| Pharmacogenomics | 7 | 7.70 | 129K | 0.35 | 0.32 | 3.60 | 4.94 |
| Diseases | 12 | 53.32 | 418K | 31.81 | 0.98 | N/A | N/A |

Table 2: Labeling function (LF) summary statistics, sizes of generated training sets $S_{\lambda \neq 0}$ (only counting non-zero labels), and relative F1 score improvement over baseline IRT methods for hand-tuned (HT) and LSTM-generated (LSTM) feature sets.

**Automatically-generated Features**   We additionally compare both hand-tuned and automatically-generated features, where the latter are learned via an LSTM recurrent neural network (RNN) [14]. Conventional wisdom states that deep learning methods such as RNNs are prone to overfitting to the biases of the imperfect rules used for programmatic supervision. In our experiments, however, we find that using data programming to denoise the labels can mitigate this issue, and we report a 9.79 point boost to precision and a 3.12 point F1 score improvement on the benchmark 2014 TAC-KBP (News) task, over the baseline *if-then-return* approach. Additionally for comparison, our approach is a 5.98 point F1 score improvement over a state-of-the-art LSTM approach [32].

**Usability Study**   One of our hopes is that a user without expertise in ML will be more productive iterating on labeling functions than on features. To test this, we arranged a hackathon involving a handful of bioinformatics researchers, using our open-source information extraction framework Snorkel[4] (formerly DDLite). Their goal was to build a disease tagging system which is a common and important challenge in the bioinformatics domain [11]. The hackathon participants did not have access to a labeled training set nor did they perform any feature engineering. The entire effort was restricted to iterative labeling function development and the setup of candidates to be classified. In under eight hours, they had created a training set that led to a model which scored within 10 points of F1 of the supervised baseline; the gap was mainly due to recall issue in the candidate extraction phase. This suggests data programming may be a promising way to build high quality extractors, quickly.

# 6   Conclusion and Future Work

We introduced data programming, a new approach to generating large labeled training sets. We demonstrated that our approach can be used with automatic feature generation techniques to achieve high quality results. We also provided anecdotal evidence that our methods may be easier for domain experts to use. We hope to explore the limits of our approach on other machine learning tasks that have been held back by the lack of high-quality supervised datasets, including those in other domains such imaging and structured prediction.

**Acknowledgements**   Thanks to Theodoros Rekatsinas, Manas Joglekar, Henry Ehrenberg, Jason Fries, Percy Liang, the `DeepDive` and `DDLite` users and many others for their helpful conversations. The authors acknowledge the support of: DARPA FA8750-12-2-0335; NSF IIS-1247701; NSFCCF-1111943; DOE 108845; NSF CCF-1337375; DARPA FA8750-13-2-0039; NSF IIS-1353606;ONR N000141210041 and N000141310129; NIH U54EB020405; DARPA's SIMPLEX program; Oracle; NVIDIA; Huawei; SAP Labs; Sloan Research Fellowship; Moore Foundation; American Family Insurance; Google; and Toshiba. The views and conclusions expressed in this material are those of the authors and should not be interpreted as necessarily representing the official policies or endorsements, either expressed or implied, of DARPA, AFRL, NSF, ONR, NIH, or the U.S. Government.

## Footnotes

[1] http://www.spacemachine.net/views/2016/3/datasets-over-algorithms

[2]http://www.nist.gov/tac/2014/KBP/

[3]https://github.com/HazyResearch/dd-genomics

[4]`snorkel.stanford.edu`

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
