[Supplementary Material]

# Data Programming:
# Creating Large Training Sets, Quickly

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

[5]`http://deepdive.stanford.edu`

[6]`http://snorkel.stanford.edu`

[7]`https://github.com/HazyResearch/treedlib`

[8]`http://deeplearning.net/software/theano/`

[9]stanfordnlp.github.io/CoreNLP/

[10]https://github.com/HazyResearch/dd-genomics

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

# A  General Theoretical Results

In this section, we will state the full form of the theoretical results we alluded to in the body of the paper. First, we restate, in long form, our setup and assumptions.

We assume that, for some function $h : \{-1, 0, 1\}^m \times \{-1, 1\} \mapsto \{-1, 0, 1\}^M$ of *sufficient statistics*, we are concerned with learning distributions, over the set $\Omega = \{-1, 0, 1\}^m \times \{-1, 1\}$, of the form

$$\pi_\theta(\Lambda, Y) = \frac{1}{Z_\theta} \exp(\theta^T h(\Lambda, Y)), \tag{10}$$

where $\theta \in \mathbb{R}^M$ is a parameter, and $Z_\theta$ is the partition function that makes this a distribution. We assume that we are given, i.e. can derive from the data programming specification, some set $\Theta$ of *feasible parameters*. This set must have the following two properties.

First, for any $\theta \in \Theta$, learning the parameter $\theta$ from (full) samples from $\pi_\theta$ is possible, at least in some sense. More specifically, there exists an unbiased estimator $\hat\theta$ that is a function of some number $D$ samples from $\pi_\theta$ (and is unbiased for all $\theta \in \Theta$) such that, for all $\theta \in \Theta$ and for some $c > 0$,

$$\mathbf{Cov}\left(\hat\theta\right) \le \frac{I}{2cD}. \tag{11}$$

Second, for any $\theta_1, \theta_2 \in \Theta$,

$$\mathbf{E}_{(\lambda_2, y_2) \sim \pi_{\theta_2}} \left[ \mathbf{Var}_{(\lambda_1, y_1) \sim \pi_{\theta_1}} (y_1 | \lambda_1 = \lambda_2) \right] \le \frac{c}{M}. \tag{12}$$

That is, we'll always be reasonably certain in our guess for the value of $y$, even if we are totally wrong about the true parameter $\theta$.

On the other hand, we are also concerned with a distribution $\pi^*$ which ranges over the set $\mathcal{X} \times \{-1, 1\}$, and represents the distribution of training and test examples we are using to learn. These objects are associated with a labeling function $\lambda : \mathcal{X} \mapsto \{-1, 0, 1\}^m$ and a feature function $f : \mathcal{X} \mapsto \mathbb{R}^n$. We make three assumptions about this distribution. First, we assume that, given $(x, y) \sim \pi^*$, the class label $y$ is independent of the features $f(x)$ given the labels $\lambda(x)$. That is,

$$(x, y) \sim \pi^* \Rightarrow y \perp f(x) \mid \lambda(x). \tag{13}$$

Second, we assume that we can describe the relationship between $\lambda(x)$ and $y$ in terms of our family in (10) above. That is, for some parameter $\theta^* \in \Theta$,

$$\mathbf{P}_{(x,y) \sim \pi^*} (\lambda(x) = \Lambda, \ y = Y) = \pi_{\theta^*}(\Lambda, Y). \tag{14}$$

Third, we assume that the features themselves are bounded; for all $x \in \mathcal{X}$,

$$\|f(x)\| \le 1. \tag{15}$$

Our goal is twofold. First, we want to recover some estimate $\hat\theta$ of the true parameter $\theta^*$. Second, we want to produce a parameter $\hat w$ that minimizes the regularized logistic loss

$$l(w) = \mathbf{E}_{(x,y) \sim \pi^*} \left[ \log(1 + \exp(-w^T f(x) y)) \right] + \rho \|w\|^2 .$$

We actually accomplish this by minimizing a noise-aware loss function, given our recovered parameter $\hat\theta$,

$$l_{\hat\theta}(w) = \mathbf{E}_{(\bar x, \bar y) \sim \pi^*} \left[ \mathbf{E}_{(\Lambda, Y) \sim \pi_{\hat\theta}} \left[ \log(1 + \exp(-w^T f(\bar x) Y)) \big| \Lambda = \lambda(\bar x) \right] \right] + \rho \|w\|^2 .$$

In fact we can't even minimize this; rather, we will be minimizing the empirical noise-aware loss function, which is only this in expectation. Since the analysis of logistic regression is not itself interesting, we assume that we are able to run some algorithm that produces an estimate $\hat w$ which satisfies, for some $\chi > 0$,

$$\mathbf{E}\left[ l_{\hat\theta}(\hat w) - \min_w l_{\hat\theta}(w) \Big| \hat\theta \right] \le \chi. \tag{16}$$

The algorithm chosen can be anything, but in practice, we use stochastic gradient descent.

We learn $\hat\theta$ and $\hat w$ by running the following algorithm.

Under these assumptions, we are able to prove the following theorem about the behavior of Algorithm 1.

---

**Algorithm 1** Data Programming

---

**Require:** Step size $\eta$, dataset $S \subset X$, and initial parameter $\theta_0 \in \Theta$.

  $\theta \to \theta_0$
  **for all** $x \in S$ **do**
    Independently sample $(\Lambda, Y)$ from $\pi_\theta$, and $(\bar{\Lambda}, \bar{Y})$ from $\pi_\theta$ conditionally given $\Lambda = \lambda(x)$.
    $\theta \leftarrow \theta + \eta(h(\Lambda, Y) - h(\bar{\Lambda}, \bar{Y}))$.
    $\theta = P_\Theta(\theta)$     ▷ *Here, $P_\Theta$ denotes orthogonal projection onto $\Theta$.*
  **end for**
  Compute $\hat{w}$ using the algorithm described in (15)
  **return** $(\theta, \hat{w})$.

---

**Theorem A.1.** Suppose that we run Algorithm 1 on a data programming specification that satisfies conditions (11), (12), (13), (14), (15), and (16). Suppose further that, for some parameter $\epsilon > 0$, we use step size

$$\eta = \frac{c\epsilon^2}{4}$$

and our dataset is of a size that satisfies

$$|S| = \frac{2}{c^2\epsilon^2} \log\left(\frac{2\|\theta_0 - \theta^*\|^2}{\epsilon}\right).$$

Then, we can bound the expected parameter error with

$$\mathbf{E}\left[\left\|\hat{\theta} - \theta^*\right\|^2\right] \leq \epsilon^2 M$$

and the expected risk with

$$\mathbf{E}\left[l(\hat{w}) - \min_w l(w)\right] \leq \chi + \frac{c\epsilon}{2\rho}.$$

This theorem's conclusions and assumptions can readily be seen to be identical to those of Theorem 2 in the main body of the paper, except that they apply to the slightly more general case of arbitrary $h$, rather than $h$ of the explicit form described in the body. Therefore, in order to prove Theorem 2, it suffices to prove Theorem A.1, which we will do in Section C.

## B   Theoretical Results for Independent Model

For the independent model, we can obtain a more specific version of Theorem A.1. In the independent model, the variables are, as before, $\Lambda \in \{-1, 0, 1\}^m$ and $Y \in \{-1, 1\}$. The sufficient statistics are $\Lambda_i Y$ and $\Lambda_i^2$.

To produce results that make intuitive sense, we also define the alternate parameterization

$$\mathbf{P}_\pi(\Lambda_i|Y) = \begin{cases} \beta_i \frac{1+\gamma_i}{2} & \Lambda_i = Y \\ (1 - \beta_i) & \Lambda = 0 \\ \beta_i \frac{1-\gamma_i}{2} & \Lambda_i = -Y \end{cases}.$$

In comparison to the parameters used in the body of the paper, we have

$$\alpha_i = \frac{1 + \gamma_i}{2}.$$

Now, we are concerned with models that are feasible. For a model to be feasible (i.e. for $\theta \in \Theta$), we require that it satisfy, for some constants $\gamma_{\min} > 0$, $\gamma_{\max} > 0$, and $\beta_{\min}$,

$$\gamma_{\min} \leq \gamma_i \leq \gamma_{\max} \qquad \beta_{\min} \leq \beta_i \leq \frac{1}{2}.$$

For $0 \leq \beta \leq 1$ and $-1 \leq \gamma \leq 1$.

For this model, we can prove the following corollary to Theorem A.1

**Corollary B.1.** *Suppose that we run Algorithm 1 on an independent data programming specification that satisfies conditions (13), (14), (15), and (16). Furthermore, assume that the number of labeling functions we use satisfies*

$$m \geq \frac{9.34 \, \text{artanh}(\gamma_{\max})}{(\gamma\beta)_{\min}\gamma_{\min}^2} \log\left(\frac{24m}{\beta_{\min}}\right).$$

*Suppose further that, for some parameter $\epsilon > 0$, we use step size*

$$\eta = \frac{\beta_{\min}\epsilon^2}{16}$$

*and our dataset is of a size that satisfies*

$$|S| = \frac{32}{\beta_{\min}^2 \epsilon^2} \log\left(\frac{2\|\theta_0 - \theta^*\|^2}{\epsilon}\right).$$

*Then, we can bound the expected parameter error with*

$$\mathbf{E}\left[\|\hat{\theta} - \theta^*\|^2\right] \leq \epsilon^2 M$$

*and the expected risk with*

$$\mathbf{E}\left[l(\hat{w}) - \min_w l(w)\right] \leq \chi + \frac{\beta_{\min}\epsilon}{8\rho}.$$

We can see that if, as stated in the body of the paper, $\beta_i \geq 0.3$ and $0.8 \leq \alpha_i \leq 0.9$ (which is equivalent to $0.6 \leq \gamma_i \leq 0.8$), then

$$2000 \geq 1896.13 = \frac{9.34 \, \text{artanh}(0.8)}{0.3 \cdot 0.6^3} \log\left(\frac{24 \cdot 2000}{0.3}\right).$$

This means that, as stated in the paper, $m = 2000$ is sufficient for this corollary to hold with

$$|S| = \frac{32}{0.3^2 \cdot \epsilon^2} \log\left(\frac{2m(\text{artanh}(0.8) - \text{artanh}(0.6))^2}{\epsilon}\right) = \frac{356}{\epsilon^2} \log\left(\frac{m}{3\epsilon}\right).$$

Thus, proving Corollary B.1 is sufficient to prove Theorem 1 from the body of the paper. We will prove Corollary B.1 in Section E

## C   Proof of Theorem A.1

First, we state some lemmas that will be useful in the proof to come.

**Lemma D.1.** *Given a family of maximum-entropy distributions*

$$\pi_\theta(x) = \frac{1}{Z_\theta} \exp(\theta^T h(x)),$$

*for some function of sufficient statistics $h : \Omega \mapsto \mathbb{R}^M$, if we let $J : \mathbb{R}^M \mapsto \mathbb{R}$ be the maximum log-likelihood objective for some event $A \subseteq \Omega$,*

$$J(\theta) = \log \mathbf{P}_{x \sim \pi_\theta}(x \in A),$$

*then its gradient is*

$$\nabla J(\theta) = \mathbf{E}_{x \sim \pi_\theta}[h(x)|x \in A] - \mathbf{E}_{x \sim \pi_\theta}[h(x)]$$

*and its Hessian is*

$$\nabla^2 J(\theta) = \mathbf{Cov}_{x \sim \pi_\theta}(h(x)|x \in A) - \mathbf{Cov}_{x \sim \pi_\theta}(h(x)).$$

**Lemma D.2.** *Suppose that we are looking at a distribution from a data programming label model. That is, our maximum-entropy distribution can now be written in terms of two variables, the labeling function values $\lambda \in \{-1, 0, 1\}$ and the class $y \in \{-1, 1\}$, as*

$$\pi_\theta(\lambda, y) = \frac{1}{Z_\theta} \exp(\theta^T h(\lambda, y)),$$

*where we assume without loss of generality that for some M, $h(\lambda, y) \in \mathbb{R}^M$ and $\|h(\lambda, y)\|_\infty \le 1$. If we let $J : \mathbb{R}^M \mapsto \mathbb{R}$ be the maximum expected log-likelihood objective, under another distribution $\pi^*$, for the event associated with the observed labeling function values $\lambda$,*

$$J(\theta) = \mathbf{E}_{(\lambda^*, y^*) \sim \pi^*} \left[ \log \mathbf{P}_{(\lambda, y) \sim \pi_\theta} (\lambda = \lambda^*) \right],$$

*then its Hessian can be bounded with*

$$\nabla^2 J(\theta) \le M I \mathbf{E}_{(\lambda^*, y^*) \sim \pi^*} \left[ \mathbf{Var}_{(\lambda, y) \sim \pi_\theta} (y | \lambda = \lambda^*) \right] - \mathcal{I}(\theta),$$

*where $\mathcal{I}(\theta)$ is the Fisher information.*

**Lemma D.3.** *Suppose that we are looking at a data programming distribution, as described in the text of Lemma D.2. Suppose further that we are concerned with some feasible set of parameters $\Theta \subset \mathbb{R}^M$, such that the any model with parameters in this space satisfies the following two conditions.*

*First, for any $\theta \in \Theta$, learning the parameter $\theta$ from (full) samples from $\pi_\theta$ is possible, at least in some sense. More specifically, there exists an unbiased estimator $\hat\theta$ that is a function of some number D samples from $\pi_\theta$ (and is unbiased for all $\theta \in \Theta$) such that, for all $\theta \in \Theta$ and for some $c > 0$,*

$$\mathbf{Cov}\left(\hat\theta\right) \le \frac{I}{2cD}.$$

*Second, for any $\theta, \theta^* \in \Theta$,*

$$\mathbf{E}_{(\lambda^*, y^*) \sim \pi^*} \left[ \mathbf{Var}_{(\lambda, y) \sim \pi_\theta} (y | \lambda = \lambda^*) \right] \le \frac{c}{M}.$$

*That is, we'll always be reasonably certain in our guess for the value of y, even if we are totally wrong about the true parameter $\theta^*$.*

*Under these conditions, the function J is strongly concave on $\Theta$ with parameter of strong convexity c.*

**Lemma D.4.** *Suppose that we are looking at a data programming maximum likelihood estimation problem, as described in the text of Lemma D.2. Suppose further that the objective function J is strongly concave with parameter $c > 0$.*

*If we run stochastic gradient descent on objective J, using unbiased samples from a true distribution $\pi_{\theta^*}$, where $\theta^* \in \Theta$, then if we use step size*

$$\eta = \frac{c\epsilon^2}{4}$$

*and run (using a fresh sample at each iteration) for T steps, where*

$$T = \frac{2}{c^2 \epsilon^2} \log \left( \frac{2 \|\theta_0 - \theta^*\|^2}{\epsilon} \right)$$

*then we can bound the expected parameter estimation error with*

$$\mathbf{E}\left[ \left\| \hat\theta - \theta^* \right\|^2 \right] \le \epsilon^2 M.$$

**Lemma D.5.** *Assume in our model that, without loss of generality, $\|f(x)\| \le 1$ for all x, and that in our true model $\pi^*$, the class y is independent of the features $f(x)$ given the labels $\lambda(x)$.*

*Suppose that we now want to solve the expected loss minimization problem wherein we minimize the objective*

$$l(w) = \mathbf{E}_{(x, y) \sim \pi^*} \left[ \log(1 + \exp(-w^T f(x) y)) \right] + \rho \|w\|^2.$$

*We actually accomplish this by minimizing our noise-aware loss function, given our chosen parameter $\hat\theta$,*

$$l_{\hat\theta}(w) = \mathbf{E}_{(\bar{x}, \bar{y}) \sim \pi^*} \left[ \mathbf{E}_{(\Lambda, Y) \sim \pi_{\hat\theta}} \left[ \log(1 + \exp(-w^T f(\bar{x}) Y)) \big| \Lambda = \lambda(\bar{x}) \right] \right] + \rho \|w\|^2.$$

*In fact we can't even minimize this; rather, we will be minimizing the empirical noise-aware loss function, which is only this in expectation. Suppose that doing so produces an estimate $\hat{w}$ which satisfies, for some $\chi > 0$,*

$$\mathbf{E}\left[ l_{\hat\theta}(\hat{w}) - \min_w l_{\hat\theta}(w) \Big| \hat\theta \right] \le \chi.$$

*(Here, the expectation is taken with respect to only the random variable $\hat{w}$.) Then, we can bound the expected risk with*

$$\mathbf{E}\left[ l(\hat{w}) - \min_w l(w) \right] \le \chi + \frac{c\epsilon}{2\rho}.$$

Now, we restate and prove our main theorem.

**Theorem A.1.** Suppose that we run Algorithm 1 on a data programming specification that satisfies conditions (11), (12), (13), (14), (15), and (16). Suppose further that, for some parameter $\epsilon > 0$, we use step size

$$\eta = \frac{c\epsilon^2}{4}$$

and our dataset is of a size that satisfies

$$|S| = \frac{2}{c^2\epsilon^2} \log\left(\frac{2\|\theta_0 - \theta^*\|^2}{\epsilon}\right).$$

Then, we can bound the expected parameter error with

$$\mathbf{E}\left[\|\hat{\theta} - \theta^*\|^2\right] \leq \epsilon^2 M$$

and the expected risk with

$$\mathbf{E}\left[l(\hat{w}) - \min_w l(w)\right] \leq \chi + \frac{c\epsilon}{2\rho}.$$

*Proof.* The bounds on the expected parameter estimation error follow directly from Lemma D.4, and the remainder of the theorem follows directly from Lemma D.5. $\square$

## D  Proofs of Lemmas

**Lemma D.1.** *Given a family of maximum-entropy distributions*

$$\pi_\theta(x) = \frac{1}{Z_\theta} \exp(\theta^T h(x)),$$

*for some function of sufficient statistics $h : \Omega \mapsto \mathbb{R}^M$, if we let $J : \mathbb{R}^M \mapsto \mathbb{R}$ be the maximum log-likelihood objective for some event $A \subseteq \Omega$,*

$$J(\theta) = \log \mathbf{P}_{x \sim \pi_\theta}(x \in A),$$

*then its gradient is*

$$\nabla J(\theta) = \mathbf{E}_{x \sim \pi_\theta}[h(x)|x \in A] - \mathbf{E}_{x \sim \pi_\theta}[h(x)]$$

*and its Hessian is*

$$\nabla^2 J(\theta) = \mathbf{Cov}_{x \sim \pi_\theta}(h(x)|x \in A) - \mathbf{Cov}_{x \sim \pi_\theta}(h(x)).$$

*Proof.* For the gradient,

$$\begin{aligned}
\nabla J(\theta) &= \nabla \log \mathbf{P}_{\pi_\theta}(A) \\
&= \nabla \log\left(\frac{\sum_{x \in A} \exp(\theta^T h(x))}{\sum_{x \in \Omega} \exp(\theta^T h(x))}\right) \\
&= \nabla \log\left(\sum_{x \in A} \exp(\theta^T h(x))\right) - \nabla \log\left(\sum_{x \in \Omega} \exp(\theta^T h(x))\right) \\
&= \frac{\sum_{x \in A} h(x) \exp(\theta^T h(x))}{\sum_{x \in A} \exp(\theta^T h(x))} - \frac{\sum_{x \in \Omega} h(x) \exp(\theta^T h(x))}{\sum_{x \in \Omega} \exp(\theta^T h(x))} \\
&= \mathbf{E}_{x \sim \pi_\theta}[h(x)|x \in A] - \mathbf{E}_{x \sim \pi_\theta}[h(x)].
\end{aligned}$$

And for the Hessian,

$$\nabla^2 J(\theta) = \nabla \frac{\sum_{x \in A} h(x) \exp(\theta^T h(x))}{\sum_{x \in A} \exp(\theta^T h(x))} - \nabla \frac{\sum_{x \in \Omega} h(x) \exp(\theta^T h(x))}{\sum_{x \in \Omega} \exp(\theta^T h(x))}$$

$$= \frac{\sum_{x \in A} h(x) h(x)^T \exp(\theta^T h(x))}{\sum_{x \in A} \exp(\theta^T h(x))} - \frac{\left(\sum_{x \in A} h(x) \exp(\theta^T h(x))\right)\left(\sum_{x \in A} h(x) \exp(\theta^T h(x))\right)^T}{\left(\sum_{x \in A} \exp(\theta^T h(x))\right)^2}$$

$$- \left( \frac{\sum_{x \in \Omega} h(x) h(x)^T \exp(\theta^T h(x))}{\sum_{x \in \Omega} \exp(\theta^T h(x))} - \frac{\left(\sum_{x \in \Omega} h(x) \exp(\theta^T h(x))\right)\left(\sum_{x \in \Omega} h(x) \exp(\theta^T h(x))\right)^T}{\left(\sum_{x \in \Omega} \exp(\theta^T h(x))\right)^2} \right)$$

$$= \mathbf{E}_{x \sim \pi_\theta}\left[ h(x) h(x)^T \big| x \in A \right] - \mathbf{E}_{x \sim \pi_\theta}\left[ h(x) | x \in A \right] \mathbf{E}_{x \sim \pi_\theta}\left[ h(x) | x \in A \right]^T$$

$$- \left( \mathbf{E}_{x \sim \pi_\theta}\left[ h(x) h(x)^T \right] - \mathbf{E}_{x \sim \pi_\theta}\left[ h(x) \right] \mathbf{E}_{x \sim \pi_\theta}\left[ h(x) \right]^T \right)$$

$$= \mathbf{Cov}_{x \sim \pi_\theta}\left( h(x) | x \in A \right) - \mathbf{Cov}_{x \sim \pi_\theta}\left( h(x) \right).$$

$\square$

**Lemma D.2.** *Suppose that we are looking at a distribution from a data programming label model. That is, our maximum-entropy distribution can now be written in terms of two variables, the labeling function values $\lambda \in \{-1, 0, 1\}$ and the class $y \in \{-1, 1\}$, as*

$$\pi_\theta(\lambda, y) = \frac{1}{Z_\theta} \exp(\theta^T h(\lambda, y)),$$

*where we assume without loss of generality that for some $M$, $h(\lambda, y) \in \mathbb{R}^M$ and $\|h(\lambda, y)\|_\infty \leq 1$. If we let $J : \mathbb{R}^M \mapsto \mathbb{R}$ be the maximum expected log-likelihood objective, under another distribution $\pi^*$, for the event associated with the observed labeling function values $\lambda$,*

$$J(\theta) = \mathbf{E}_{(\lambda^*, y^*) \sim \pi^*}\left[ \log \mathbf{P}_{(\lambda, y) \sim \pi_\theta}\left( \lambda = \lambda^* \right) \right],$$

*then its Hessian can be bounded with*

$$\nabla^2 J(\theta) \leq M I \mathbf{E}_{(\lambda^*, y^*) \sim \pi^*}\left[ \mathbf{Var}_{(\lambda, y) \sim \pi_\theta}\left( y | \lambda = \lambda^* \right) \right] - \mathcal{I}(\theta),$$

*where $\mathcal{I}(\theta)$ is the Fisher information.*

*Proof.* From the result of Lemma D.1, we have that

$$\nabla^2 J(\theta) = \mathbf{E}_{(\lambda^*, y^*) \sim \pi^*}\left[ \mathbf{Cov}_{(\lambda, y) \sim \pi_\theta}\left( h(\lambda, y) | \lambda = \lambda^* \right) \right] - \mathbf{Cov}_{(\lambda, y) \sim \pi_\theta}\left( h(\lambda, y) \right). \tag{17}$$

We start byu defining $h_0(\lambda)$ and $h_1(\lambda)$ such that

$$h(\lambda, y) = h(\lambda, 1) \frac{1 + y}{2} + h(\lambda, -1) \frac{1 - y}{2} = \frac{h(\lambda, 1) + h(\lambda, -1)}{2} + y \frac{h(\lambda, 1) - h(\lambda, -1)}{2} = h_0(\lambda) + y h_1(\lambda).$$

This allows us to reduce (17) to

$$\nabla^2 J(\theta) = \mathbf{E}_{(\lambda^*, y^*) \sim \pi^*}\left[ h_1(\lambda^*) h_1(\lambda^*)^T \mathbf{Var}_{(\lambda, y) \sim \pi_\theta}\left( y | \lambda = \lambda^* \right) \right] - \mathbf{Cov}_{(\lambda, y) \sim \pi_\theta}\left( h(\lambda, y) \right).$$

On the other hand, the Fisher information of this model at $\theta$ is

$$\mathcal{I}(\theta) = \mathbf{E}\left[ \left( \nabla_\theta \log \pi_\theta(x) \right)^2 \right]$$

$$= \mathbf{E}\left[ \left( \nabla_\theta \log \left( \frac{\exp(\theta^T h(x))}{\sum_{z \in \Omega} \exp(\theta^T h(z))} \right) \right)^2 \right]$$

$$= \mathbf{E}\left[ \left( \nabla_\theta \log \left( \exp(\theta^T h(x)) \right) - \nabla_\theta \log \left( \sum_{z \in \Omega} \exp(\theta^T h(z)) \right) \right)^2 \right]$$

$$= \mathbf{E}\left[ \left( h(x) - \frac{\sum_{z \in \Omega} h(z) \exp(\theta^T h(z))}{\sum_{z \in \Omega} \exp(\theta^T h(z))} \right)^2 \right]$$

$$= \mathbf{E}\left[ \left( h(x) - \mathbf{E}\left[ h(z) \right] \right)^2 \right]$$

$$= \mathbf{Cov}\left( h(x) \right).$$

Therefore, we can write the second derivative of $J$ as

$$\nabla^2 J(\theta) = \mathbf{E}_{(\lambda^*,y^*) \sim \pi^*} \left[ h_1(\lambda^*) h_1(\lambda^*)^T \mathbf{Var}_{(\lambda,y) \sim \pi_\theta} (y|\lambda = \lambda^*) \right] - \mathcal{I}(\theta).$$

If we apply the fact that

$$h_1(\lambda^*) h_1(\lambda^*)^T \preceq I \|h_1(\lambda^*)\|^2 \leq MI \|h_1(\lambda^*)\|_\infty^2 \leq MI,$$

then we can reduce this to

$$\nabla^2 J(\theta) \preceq MI \mathbf{E}_{(\lambda^*,y^*) \sim \pi^*} \left[ \mathbf{Var}_{(\lambda,y) \sim \pi_\theta} (y|\lambda = \lambda^*) \right] - \mathcal{I}(\theta).$$

This is the desired result.                                                   □

**Lemma D.3.** *Suppose that we are looking at a data programming distribution, as described in the text of Lemma D.2. Suppose further that we are concerned with some feasible set of parameters $\Theta \subset \mathbb{R}^M$, such that the any model with parameters in this space satisfies the following two conditions.*

*First, for any $\theta \in \Theta$, learning the parameter $\theta$ from (full) samples from $\pi_\theta$ is possible, at least in some sense. More specifically, there exists an unbiased estimator $\hat\theta$ that is a function of some number $D$ samples from $\pi_\theta$ (and is unbiased for all $\theta \in \Theta$) such that, for all $\theta \in \Theta$ and for some $c > 0$,*

$$\mathbf{Cov} \left( \hat\theta \right) \preceq \frac{I}{2cD}.$$

*Second, for any $\theta, \theta^* \in \Theta$,*

$$\mathbf{E}_{(\lambda^*,y^*) \sim \pi^*} \left[ \mathbf{Var}_{(\lambda,y) \sim \pi_\theta} (y|\lambda = \lambda^*) \right] \leq \frac{c}{M}.$$

*That is, we'll always be reasonably certain in our guess for the value of y, even if we are totally wrong about the true parameter $\theta^*$.*

*Under these conditions, the function $J$ is strongly concave on $\Theta$ with parameter of strong convexity c.*

*Proof.* From the Cramér-Rao bound, we know in general that the variance of any unbiased estimator is bounded by the reciprocal of the Fisher information

$$\mathbf{Cov} \left( \hat\theta \right) \succeq (\mathcal{I}(\theta))^{-1} .$$

Since for the estimator described in the lemma statement, we have $D$ independent samples from the distribution, it follows that the Fisher information of this experiment is $D$ times the Fisher information of a single sample. Combining this with the bound in the lemma statement on the covariance, we get

$$\frac{I}{2cD} \succeq \mathbf{Cov} \left( \hat\theta \right) \succeq (D\mathcal{I}(\theta))^{-1} .$$

It follows that

$$\mathcal{I}(\theta) \succeq 2cI.$$

On the other hand, also from the lemma statement, we can conclude that

$$MI \mathbf{E}_{(\lambda^*,y^*) \sim \pi^*} \left[ \mathbf{Var}_{(\lambda,y) \sim \pi_\theta} (y|\lambda = \lambda^*) \right] \preceq cI.$$

Therefore, for all $\theta \in \Theta$,

$$\nabla^2 J(\theta) \preceq MI \mathbf{E}_{(\lambda^*,y^*) \sim \pi^*} \left[ \mathbf{Var}_{(\lambda,y) \sim \pi_\theta} (y|\lambda = \lambda^*) \right] - \mathcal{I}(\theta) \preceq -cI.$$

This implies that $J$ is strongly concave over $\Theta$, with constant $c$, as desired.                □

**Lemma D.4.** *Suppose that we are looking at a data programming maximum likelihood estimation problem, as described in the text of Lemma D.2. Suppose further that the objective function $J$ is strongly concave with parameter $c > 0$.*

*If we run stochastic gradient descent on objective $J$, using unbiased samples from a true distribution $\pi_{\theta^*}$, where $\theta^* \in \Theta$, then if we use step size*

$$\eta = \frac{c\epsilon^2}{4}$$

*and run (using a fresh sample at each iteration) for $T$ steps, where*

$$T = \frac{2}{c^2 \epsilon^2} \log\left(\frac{2\|\theta_0 - \theta^*\|^2}{\epsilon}\right)$$

*then we can bound the expected parameter estimation error with*

$$\mathbf{E}\left[\|\hat{\theta} - \theta^*\|^2\right] \leq \epsilon^2 M.$$

*Proof.* First, we note that, in the proof to follow, we can ignore the projection onto the feasible set $\Theta$, since this projection always takes us closer to the optimum $\theta^*$.

If we track the expected distance to the optimum $\theta^*$, then at the next timestep,

$$\|\theta_{t+1} - \theta^*\|^2 = \|\theta_t - \theta^*\|^2 + 2\gamma(\theta_t - \theta^*)\nabla \tilde{J}(\theta_t) + \gamma^2 \left\|\nabla \tilde{J}_t(\theta_t)\right\|^2.$$

Since we can write our stochastic samples in the form

$$\nabla \tilde{J}_t(\theta_t) = h(\lambda_t, y_t) - h(\bar{\lambda}_t, \bar{y}_t),$$

for some samples $\lambda_t$, $y_t$, $\bar{\lambda}_t$, and $\bar{y}_t$, we can conclude that

$$\left\|\nabla \tilde{J}_t(\theta_t)\right\|^2 \leq M \left\|\nabla \tilde{J}_t(\theta_t)\right\|_\infty^2 \leq 4M.$$

Therefore, taking the expected value conditioned on the filtration,

$$\mathbf{E}\left[\|\theta_{t+1} - \theta^*\|^2 \big| \mathcal{F}_t\right] = \|\theta_t - \theta^*\|^2 + 2\gamma(\theta_t - \theta^*)\nabla J(\theta_t) + 4\gamma^2 M.$$

Since $J$ is strongly concave,

$$(\theta_t - \theta^*)\nabla J(\theta_t) \leq -c \|\theta_t - \theta^*\|^2 ;$$

and so,

$$\mathbf{E}\left[\|\theta_{t+1} - \theta^*\|^2 \big| \mathcal{F}_t\right] \leq (1 - 2\gamma c) \|\theta_t - \theta^*\|^2 + 4\gamma^2 M.$$

If we take the full expectation and subtract the fixed point from both sides,

$$\mathbf{E}\left[\|\theta_{t+1} - \theta^*\|^2\right] - \frac{2\gamma M}{c} \leq (1 - 2\gamma c)\mathbf{E}\left[\|\theta_t - \theta^*\|^2\right] + 4\gamma^2 M - \frac{2\gamma M}{c} = (1 - 2\gamma c)\left(\mathbf{E}\left[\|\theta_t - \theta^*\|^2\right] - \frac{2\gamma M}{c}\right).$$

Therefore,

$$\mathbf{E}\left[\|\theta_t - \theta^*\|^2\right] - \frac{2\gamma M}{c} \leq (1 - 2\gamma c)^t \left(\|\theta_0 - \theta^*\|^2 - \frac{2\gamma M}{c}\right),$$

and so

$$\mathbf{E}\left[\|\theta_t - \theta^*\|^2\right] \leq \exp(-2\gamma ct) \|\theta_0 - \theta^*\|^2 + \frac{2\gamma M}{c}.$$

In order to ensure that

$$\mathbf{E}\left[\|\theta_t - \theta^*\|^2\right] \leq \epsilon^2,$$

it therefore suffices to pick

$$\gamma = \frac{c\epsilon^2}{4M}$$

and

$$t = \frac{2M}{c^2 \epsilon^2} \log\left(\frac{2\|\theta_0 - \theta^*\|^2}{\epsilon}\right).$$

Substituting $\epsilon^2 \to \epsilon^2 M$ produces the desired result. $\qquad\square$

**Lemma D.5.** *Assume in our model that, without loss of generality, $\|f(x)\| \leq 1$ for all $x$, and that in our true model $\pi^*$, the class $y$ is independent of the features $f(x)$ given the labels $\lambda(x)$.*

*Suppose that we now want to solve the expected loss minimization problem wherein we minimize the objective*

$$l(w) = \mathbf{E}_{(x,y)\sim\pi^*}\left[\log(1 + \exp(-w^T f(x)y))\right] + \rho \|w\|^2.$$

*We actually accomplish this by minimizing our noise-aware loss function, given our chosen parameter* $\hat{\theta}$,

$$l_{\hat{\theta}}(w) = \mathbf{E}_{(\bar{x},\bar{y})\sim\pi^*}\left[\mathbf{E}_{(\Lambda,Y)\sim\pi_{\hat{\theta}}}\left[\log(1+\exp(-w^T f(\bar{x})Y))\big|\Lambda = \lambda(\bar{x})\right]\right] + \rho\,\|w\|^2\,.$$

*In fact we can't even minimize this; rather, we will be minimizing the empirical noise-aware loss function, which is only this in expectation. Suppose that doing so produces an estimate* $\hat{w}$ *which satisfies, for some* $\chi > 0$,

$$\mathbf{E}\left[l_{\hat{\theta}}(\hat{w}) - \min_w l_{\hat{\theta}}(w)\Big|\hat{\theta}\right] \leq \chi\,.$$

*(Here, the expectation is taken with respect to only the random variable* $\hat{w}$.*) Then, we can bound the expected risk with*

$$\mathbf{E}\left[l(\hat{w}) - \min_w l(w)\right] \leq \chi + \frac{c\epsilon}{2\rho}\,.$$

*Proof.* (To simplify the symbols in this proof, we freely use $\theta$ when we mean $\hat{\theta}$.)

The loss function we want to minimize is, in expectation,

$$l(w) = \mathbf{E}_{(x,y)\sim\pi^*}\left[\log(1+\exp(-w^T f(x)y))\right] + \rho\,\|w\|^2\,.$$

By the law of total expectation,

$$l(w) = \mathbf{E}_{(\bar{x},\bar{y})\sim\pi^*}\left[\mathbf{E}_{(x,y)\sim\pi^*}\left[\log(1+\exp(-w^T f(\bar{x})y))\big|x = \bar{x}\right]\right] + \rho\,\|w\|^2\,,$$

and by our conditional independence assumption,

$$l(w) = \mathbf{E}_{(\bar{x},\bar{y})\sim\pi^*}\left[\mathbf{E}_{(x,y)\sim\pi^*}\left[\log(1+\exp(-w^T f(\bar{x})y))\big|\lambda(x) = \lambda(\bar{x})\right]\right] + \rho\,\|w\|^2\,.$$

Since we know from our assumptions that, for the optimum parameter $\theta^*$,

$$\mathbf{P}_{(x,y)\sim\pi^*}\left(\lambda(x) = \Lambda, y = Y\right) = \mathbf{P}_{(\lambda,y)\sim\pi_{\theta^*}}\left(\lambda = \Lambda, y = Y\right),$$

we can rewrite this as

$$l(w) = \mathbf{E}_{(\bar{x},\bar{y})\sim\pi^*}\left[\mathbf{E}_{(\Lambda,Y)\sim\pi_{\theta^*}}\left[\log(1+\exp(-w^T f(\bar{x})Y))\big|\Lambda = \lambda(\bar{x})\right]\right] + \rho\,\|w\|^2\,.$$

On the other hand, if we are minimizing the model we got from the previous step, we will be actually minimizing

$$l_{\theta}(w) = \mathbf{E}_{(\bar{x},\bar{y})\sim\pi^*}\left[\mathbf{E}_{(\Lambda,Y)\sim\pi_{\theta}}\left[\log(1+\exp(-w^T f(\bar{x})Y))\big|\Lambda = \lambda(\bar{x})\right]\right] + \rho\,\|w\|^2\,.$$

We can reduce this further by noticing that

$$\mathbf{E}_{(\Lambda,Y)\sim\pi_{\theta}}\left[\log(1+\exp(-w^T f(\bar{x})Y))\big|\Lambda = \lambda(\bar{x})\right]$$

$$= \mathbf{E}_{(\Lambda,Y)\sim\pi_{\theta}}\left[\log(1+\exp(-w^T f(\bar{x})))\frac{1+Y}{2} + \log(1+\exp(w^T f(\bar{x})))\frac{1-Y}{2}\bigg|\Lambda = \lambda(\bar{x})\right]$$

$$= \frac{\log(1+\exp(-w^T f(\bar{x}))) + \log(1+\exp(w^T f(\bar{x})))}{2}$$

$$\quad + \frac{\log(1+\exp(-w^T f(\bar{x}))) - \log(1+\exp(w^T f(\bar{x})))}{2}\mathbf{E}_{(\Lambda,Y)\sim\pi_{\theta}}\left[Y|\Lambda = \lambda(\bar{x})\right]$$

$$= \frac{\log(1+\exp(-w^T f(\bar{x}))) + \log(1+\exp(w^T f(\bar{x})))}{2}$$

$$\quad - \frac{w^T f(\bar{x})}{2}\mathbf{E}_{(\Lambda,Y)\sim\pi_{\theta}}\left[Y|\Lambda = \lambda(\bar{x})\right]\,.$$

It follows that the difference between the loss functions will be

$$|l(w) - l_{\theta}(w)| = \left|\mathbf{E}_{(\bar{x},\bar{y})\sim\pi^*}\left[\frac{w^T f(\bar{x})}{2}\left(\mathbf{E}_{(\Lambda,Y)\sim\pi_{\theta}}\left[Y|\Lambda = \lambda(\bar{x})\right] - \mathbf{E}_{(\Lambda,Y)\sim\pi_{\theta^*}}\left[Y|\Lambda = \lambda(\bar{x})\right]\right)\right]\right|\,.$$

Now, we can compute that

$$\nabla_\theta \mathbf{E}_{(\Lambda,Y)\sim\pi_\theta}[Y|\Lambda = \lambda] = \nabla_\theta \frac{\exp(\theta^T h(\lambda, 1)) - \exp(\theta^T h(\lambda, -1))}{\exp(\theta^T h(\lambda, 1)) + \exp(\theta^T h(\lambda, -1))}$$

$$= \nabla_\theta \frac{\exp(\theta^T h_1(\lambda)) - \exp(-\theta^T h_1(\lambda))}{\exp(\theta^T h_1(\lambda)) + \exp(\theta^T h_1(\lambda))}$$

$$= \nabla_\theta \tanh(\theta^T h_1(\lambda))$$

$$= h_1(\lambda)\left(1 - \tanh^2(\theta^T h_1(\lambda))\right)$$

$$= h_1(\lambda)\mathbf{Var}_{(\Lambda,Y)\sim\pi_\theta}(Y|\Lambda = \lambda).$$

It follows by the mean value theorem that for some $\psi$, a linear combination of $\theta$ and $\theta^*$,

$$|l(w) - l_\theta(w)| = \left|\mathbf{E}_{(\bar{x},\bar{y})\sim\pi^*}\left[\frac{w^T f(\bar{x})}{2}(\theta - \theta^*)^T h_1(\lambda)\mathbf{Var}_{(\Lambda,Y)\sim\pi_\psi}(Y|\Lambda = \lambda)\right]\right|.$$

Since $\Theta$ is convex, clearly $\psi \in \Theta$. From our assumption on the bound of the variance, we can conclude that

$$\mathbf{E}_{(\bar{x},\bar{y})\sim\pi^*}\left[\mathbf{Var}_{(\Lambda,Y)\sim\pi_\psi}(Y|\Lambda = \lambda)\right] \leq \frac{c}{M}.$$

By the Cauchy-Schwarz inequality,

$$|l(w) - l_\theta(w)| \leq \frac{1}{2}\left|\mathbf{E}_{(\bar{x},\bar{y})\sim\pi^*}\left[\|w\|\,\|f(\bar{x})\|\,\|\theta - \theta^*\|\,\|h_1(\lambda)\|\,\mathbf{Var}_{(\Lambda,Y)\sim\pi_\psi}(Y|\Lambda = \lambda)\right]\right|.$$

Since (by assumption) $\|f(x)\| \leq 1$ and $\|h_1(\lambda)\| \leq \sqrt{M}$,

$$|l(w) - l_\theta(w)| \leq \frac{\|w\|\,\|\theta - \theta^*\|\,\sqrt{M}}{2}\left|\mathbf{E}_{(\bar{x},\bar{y})\sim\pi^*}\left[\mathbf{Var}_{(\Lambda,Y)\sim\pi_\psi}(Y|\Lambda = \lambda)\right]\right|$$

$$\leq \frac{\|w\|\,\|\theta - \theta^*\|\,\sqrt{M}}{2}\cdot\frac{c}{M}$$

$$= \frac{c\,\|w\|\,\|\theta - \theta^*\|}{2\sqrt{M}}.$$

Now, for any $w$ that could conceivably be a solution, it must be the case that

$$\|w\| \leq \frac{1}{2\rho},$$

since otherwise the regularization term would be too large Therefore, for any possible solution $w$,

$$|l(w) - l_\theta(w)| \leq \frac{c\,\|\theta - \theta^*\|}{4\rho\sqrt{M}}.$$

Now, we apply the assumption that we are able to solve the empirical problem, producing an estimate $\hat{w}$ that satisfies

$$\mathbf{E}\left[l_\theta(\hat{w}) - l_\theta(w_\theta^*)\right] \leq \chi,$$

where $w_\theta^*$ is the true solution to

$$w_\theta^* = \arg\min_w l_\theta(w).$$

Therefore,

$$\mathbf{E}\left[l(\hat{w}) - l(w^*)\right] = \mathbf{E}\left[l_\theta(\hat{w}) - l_\theta(w_\theta^*) + l_\theta(w_\theta^*) - l_\theta(\hat{w}) + l(\hat{w}) - l(w^*)\right]$$

$$\leq \chi + \mathbf{E}\left[l_\theta(w^*) - l_\theta(\hat{w}) + l(\hat{w}) - l(w^*)\right]$$

$$\leq \chi + \mathbf{E}\left[|l_\theta(w^*) - l(w^*)| + |l_\theta(\hat{w}) - l(\hat{w})|\right]$$

$$\leq \chi + \mathbf{E}\left[\frac{c\,\|\theta - \theta^*\|}{2\rho\sqrt{M}}\right]$$

$$= \chi + \frac{c}{2\rho\sqrt{M}}\mathbf{E}\left[\|\theta - \theta^*\|\right]$$

$$\leq \chi + \frac{c}{2\rho\sqrt{M}}\sqrt{\mathbf{E}\left[\|\theta - \theta^*\|^2\right]}.$$

We can now bound this using the result of Lemma D.4, which results in

$$\mathbf{E}\left[l(\hat{w}) - l(w^*)\right] \leq \chi + \frac{c}{2\rho\sqrt{M}}\sqrt{M\epsilon^2}$$

$$= \chi + \frac{c\epsilon}{2\rho}.$$

This is the desired result. $\qquad\qquad\square$

# E  Proofs of Results for the Independent Model

To restate, in the independent model, the variables are, as before, $\Lambda \in \{-1, 0, 1\}^m$ and $Y \in \{-1, 1\}$. The sufficient statistics are $\Lambda_i Y$ and $\Lambda_i^2$. That is, for expanded parameter $\theta = (\psi, \phi)$,

$$\pi_\theta(\Lambda, Y) = \frac{1}{Z}\exp(\psi^T \Lambda Y + \phi^T \Lambda^2).$$

This can be combined with the simple assumption that $\mathbf{P}(Y) = \frac{1}{2}$ to complete a whole distribution. Using this, we can prove the following simple result about the moments of the sufficient statistics.

**Lemma E.1.** *The expected values and covariances of the sufficient statistics are, for all $i \neq j$,*

$$\mathbf{E}\left[\Lambda_i Y\right] = \beta_i \gamma_i$$

$$\mathbf{E}\left[\Lambda_i^2\right] = \beta_i$$

$$\mathbf{Var}\left(\Lambda_i Y\right) = \beta_i - \beta_i^2 \gamma_i^2$$

$$\mathbf{Var}\left(\Lambda_i^2\right) = \beta_i - \beta_i^2$$

$$\mathbf{Cov}\left(\Lambda_i Y, \Lambda_j Y\right) = 0$$

$$\mathbf{Cov}\left(\Lambda_i^2, \Lambda_j^2\right) = 0$$

$$\mathbf{Cov}\left(\Lambda_i Y, \Lambda_j^2\right) = 0.$$

We also prove the following basic lemma that relates $\psi_i$ to $\gamma_i$.

**Lemma E.2.** *It holds that*

$$\gamma_i = \tanh(\psi_i).$$

We also make the following claim about feasible models.

**Lemma E.3.** *For any feasible model, it will be the case that, for any other feasible parameter vector $\hat{\psi}$,*

$$\mathbf{P}\left(\hat{\psi}^T \Lambda Y \leq \frac{m}{2}\gamma_{\min}(\gamma\beta)_{\min}\right) \leq \exp\left(-\frac{m(\gamma\beta)_{\min}\gamma_{\min}^2}{9.34\,\mathrm{artanh}(\gamma_{\max})}\right).$$

We can also prove the following simple result about the conditional covariances

**Lemma E.4.** *The covariances of the sufficient statistics, conditioned on $\Lambda$, are for all $i \neq j$,*

$$\mathbf{Cov}\left(\Lambda_i Y, \Lambda_j Y \mid \Lambda\right) = \Lambda_i \Lambda_j \operatorname{sech}^2(\psi^T \Lambda)$$

$$\mathbf{Cov}\left(\Lambda_i^2, \Lambda_j^2 \mid \Lambda\right) = 0.$$

We can combine these two results to bound the expected variance of these conditional statistics.

**Lemma E.5.** *If $\theta$ and $\theta^*$ are two feasible models, then for any $u$,*

$$\mathbf{E}_{\theta^*}\left[\mathbf{Var}_\theta(Y \mid \Lambda)\right] \leq 3\exp\left(-\frac{m\beta_{\min}^2\gamma_{\min}^3}{8\,\mathrm{artanh}(\gamma_{\max})}\right).$$

We can now proceed to restate and prove the main corollary of Theorem A.1 that applies in the independent case.

**Corollary B.1.** *Suppose that we run Algorithm 1 on an independent data programming specification that satisfies conditions (13), (14), (15), and (16). Furthermore, assume that the number of labeling functions we use satisfies*

$$m \geq \frac{9.34 \operatorname{artanh}(\gamma_{\max})}{(\gamma\beta)_{\min}\gamma_{\min}^2} \log\left(\frac{24m}{\beta_{\min}}\right).$$

*Suppose further that, for some parameter $\epsilon > 0$, we use step size*

$$\eta = \frac{\beta_{\min}\epsilon^2}{16}$$

*and our dataset is of a size that satisfies*

$$|S| = \frac{32}{\beta_{\min}^2\epsilon^2} \log\left(\frac{2\|\theta_0 - \theta^*\|^2}{\epsilon}\right).$$

*Then, we can bound the expected parameter error with*

$$\mathbf{E}\left[\|\hat{\theta} - \theta^*\|^2\right] \leq \epsilon^2 M$$

*and the expected risk with*

$$\mathbf{E}\left[l(\hat{w}) - \min_w l(w)\right] \leq \chi + \frac{\beta_{\min}\epsilon}{8\rho}.$$

*Proof.* In order to apply Theorem A.1, we have to verify all its conditions hold in the independent case.

First, we notice that (11) is used only to bound the covariance of the sufficient statistics. From Lemma E.1, we know that these can be bounded by $\beta_i - \beta_i^2\gamma_i^2 \geq \frac{\beta_{\min}}{2}$. It follows that we can choose

$$c = \frac{\beta_{\min}}{4},$$

and we can consider (11) satisfied, for the purposes of applying the theorem.

Second, to verify (12), we can use Lemma E.5. For this to work, we need

$$3\exp\left(-\frac{m(\gamma\beta)_{\min}\gamma_{\min}^2}{9.34 \operatorname{artanh}(\gamma_{\max})}\right) \leq \frac{c}{M} = \frac{\beta_{\min}}{8m}.$$

This happens whenever the number of labeling functions satisfies

$$m \geq \frac{9.34 \operatorname{artanh}(\gamma_{\max})}{(\gamma\beta)_{\min}\gamma_{\min}^2} \log\left(\frac{24m}{\beta_{\min}}\right).$$

The remaining assumptions, (13), (14), (15), and (16), are satisfied directly by the assumptions of this corollary. So, we can apply Theorem A.1, which produces the desired result. $\square$

# F   Proofs of Independent Model Lemmas

**Lemma E.1.** *The expected values and covariances of the sufficient statistics are, for all $i \neq j$,*

$$\mathbf{E}\left[\Lambda_i Y\right] = \beta_i\gamma_i$$
$$\mathbf{E}\left[\Lambda_i^2\right] = \beta_i$$
$$\mathbf{Var}\left(\Lambda_i Y\right) = \beta_i - \beta_i^2\gamma_i^2$$
$$\mathbf{Var}\left(\Lambda_i^2\right) = \beta_i - \beta_i^2$$
$$\mathbf{Cov}\left(\Lambda_i Y, \Lambda_j Y\right) = 0$$
$$\mathbf{Cov}\left(\Lambda_i^2, \Lambda_j^2\right) = 0$$
$$\mathbf{Cov}\left(\Lambda_i Y, \Lambda_j^2\right) = 0.$$

*Proof.* We prove each of the statements in turn. For the first statement,

$$\mathbf{E}\left[\Lambda_i Y\right] = \mathbf{P}\left(\Lambda_i = Y\right) - \mathbf{P}\left(\Lambda_i = -Y\right)$$
$$= \beta_i \frac{1 + \gamma_i}{2} - \beta_i \frac{1 - \gamma_i}{2}$$
$$= \beta_i \gamma_i.$$

For the second statement,

$$\mathbf{E}\left[\Lambda_i^2\right] = \mathbf{P}\left(\Lambda = Y\right) + \mathbf{P}\left(\Lambda = -Y\right)$$
$$= \beta_i \frac{1 + \gamma_i}{2} + \beta_i \frac{1 - \gamma_i}{2}$$
$$= \beta_i.$$

For the remaining statements, we derive the second moments; converting these to an expression of the covariance is trivial. For the third statement,

$$\mathbf{E}\left[(\Lambda_i Y)^2\right] = \mathbf{E}\left[\Lambda_i^2 Y^2\right] = \mathbf{E}\left[\Lambda_i^2\right] = \beta_i.$$

For the fourth statement,

$$\mathbf{E}\left[(\Lambda_i^2)^2\right] = \mathbf{E}\left[\Lambda_i^4\right] = \mathbf{E}\left[\Lambda_i^2\right] = \beta_i.$$

For subsequent statements, we first derive that

$$\mathbf{E}\left[\Lambda_i Y | Y\right] = \beta_i \frac{1 + \gamma_i}{2} - \beta_i \frac{1 - \gamma_i}{2} = \beta_i \gamma_i$$

and

$$\mathbf{E}\left[\Lambda_i^2 | Y\right] = \beta_i \frac{1 + \gamma_i}{2} + \beta_i \frac{1 - \gamma_i}{2} = \beta_i.$$

Now, for the fifth statement,

$$\mathbf{E}\left[(\Lambda_i Y)(\Lambda_j Y)\right] = \mathbf{E}\left[\mathbf{E}\left[\Lambda_i Y | Y\right] \mathbf{E}\left[\Lambda_j Y | Y\right]\right] = \beta_i \gamma_i \beta_j \gamma_j.$$

For the sixth statement,

$$\mathbf{E}\left[(\Lambda_i^2)(\Lambda_j^2)\right] = \mathbf{E}\left[\mathbf{E}\left[\Lambda_i^2 | Y\right] \mathbf{E}\left[\Lambda_i^2 | Y\right]\right] = \beta_i \beta_j.$$

Finally, for the seventh statement,

$$\mathbf{E}\left[(\Lambda_i Y)(\Lambda_j^2)\right] = \mathbf{E}\left[\mathbf{E}\left[\Lambda_i Y | Y\right] \mathbf{E}\left[\Lambda_i^2 | Y\right]\right] = \beta_i \gamma_i \beta_j.$$

This completes the proof. □

**Lemma E.2.** *It holds that*

$$\gamma_i = \tanh(\psi_i).$$

*Proof.* From the definitions,

$$\beta_i = \frac{\exp(\psi_i + \phi_i) + \exp(-\psi_i + \phi_i)}{\exp(\psi_i + \phi_i) + \exp(-\psi_i + \phi_i) + 1}$$

and

$$\beta_i \gamma_i = \frac{\exp(\psi_i + \phi_i) - \exp(-\psi_i + \phi_i)}{\exp(\psi_i + \phi_i) + \exp(-\psi_i + \phi_i) + 1}.$$

Therefore,

$$\gamma_i = \frac{\exp(\psi_i + \phi_i) - \exp(-\psi_i + \phi_i)}{\exp(\psi_i + \phi_i) + \exp(-\psi_i + \phi_i)} = \tanh(\psi_i),$$

which is the desired result. □

**Lemma E.3.** *For any feasible model, it will be the case that, for any other feasible parameter vector* $\hat{\psi}$,

$$\mathbf{P}\left(\hat{\psi}^T \Lambda Y \leq \frac{m}{2} \gamma_{\min}(\gamma\beta)_{\min}\right) \leq \exp\left(-\frac{m(\gamma\beta)_{\min}\gamma_{\min}^2}{9.34 \operatorname{artanh}(\gamma_{\max})}\right).$$

*Proof.* We start by noticing that

$$\hat{\psi}^T \Lambda Y = \sum_{i=1}^{m} \hat{\psi}_i \Lambda_i Y.$$

Since in this model, all the $\Lambda_i Y$ are independent of each other, we can bound this sum using a concentration bound. First, we note that

$$\left| \hat{\psi}_i \Lambda_i Y \right| \leq \hat{\psi}_i.$$

Second, we note that

$$\mathbf{E}\left[ \hat{\psi}_i \Lambda_i Y \right] = \hat{\psi}_i \beta_i \gamma_i$$

and

$$\mathbf{Var}\left( \hat{\psi}_i \Lambda_i Y \right) = \hat{\psi}_i^2 \left( \beta_i - \beta_i^2 \gamma_i^2 \right)$$

but

$$\left| \hat{\psi}_i \Lambda_i Y \right| \leq \hat{\psi}_i \leq \mathrm{artanh}(\gamma_{\max}) \triangleq \hat{\psi}_{\max}$$

because, for feasible models, by definition

$$\gamma_{\min} \leq \mathrm{artanh}(\gamma_{\min}) \leq \hat{\psi}_i \leq \mathrm{artanh}(\gamma_{\max}).$$

Therefore, applying Bernstein's inequality gives us, for any $t$,

$$\mathbf{P}\left( \sum_{i=1}^{m} \hat{\psi}_i \Lambda_i Y - \sum_{i=1}^{m} \hat{\psi}_i \beta_i \gamma_i \leq -t \right) \leq \exp\left( -\frac{3t^2}{6 \sum_{i=1}^{m} \hat{\psi}_i^2 \gamma_i \beta_i \gamma_i + 2\hat{\psi}_{\max} t} \right).$$

It follows that, if we let

$$t = \frac{1}{2} \sum_{i=1}^{m} \hat{\psi}_i \beta_i \gamma_i,$$

then we get

$$\mathbf{P}\left( \sum_{i=1}^{m} \hat{\psi}_i \Lambda_i Y - \sum_{i=1}^{m} \hat{\psi}_i \beta_i \gamma_i \leq -t \right) \leq \exp\left( -\frac{3\left( \frac{1}{2} \sum_{i=1}^{m} \hat{\psi}_i \beta_i \gamma_i \right)^2}{6 \sum_{i=1}^{m} \hat{\psi}_i^2 \gamma_i \beta_i \gamma_i + 2\hat{\psi}_{\max}\left( \frac{1}{2} \sum_{i=1}^{m} \hat{\psi}_i \beta_i \gamma_i \right)} \right)$$

$$\leq \exp\left( -\frac{3 \sum_{i=1}^{m} \hat{\psi}_i \beta_i \gamma_i}{24 \gamma_{\max} \hat{\psi}_{\max} + 4\hat{\psi}_{\max}} \right)$$

$$\leq \exp\left( -\frac{3m(1 - \gamma_{\max})}{28 \hat{\psi}_{\max}} \right)$$

$$\leq \exp\left( -\frac{3\left( \sum_{i=1}^{m} \hat{\psi}_i \beta_i \gamma_i \right)^2}{24 \sum_{i=1}^{m} \hat{\psi}_i^2 \beta_i + 4\hat{\psi}_{\max}\left( \sum_{i=1}^{m} \hat{\psi}_i \beta_i \gamma_i \right)} \right)$$

$$\leq \exp\left( -\frac{3\gamma_{\min}\left( \sum_{i=1}^{m} \hat{\psi}_i \beta_i \right)\left( \sum_{i=1}^{m} \hat{\psi}_i \beta_i \gamma_i \right)}{24\hat{\psi}_{\max} \sum_{i=1}^{m} \hat{\psi}_i \beta_i + 4\hat{\psi}_{\max}\left( \sum_{i=1}^{m} \hat{\psi}_i \beta_i \right)} \right)$$

$$\leq \exp\left( -\frac{3\gamma_{\min}\left( \sum_{i=1}^{m} \hat{\psi}_i \beta_i \gamma_i \right)}{28\hat{\psi}_{\max}} \right)$$

$$\leq \exp\left( -\frac{m\gamma_{\min}^2 (\gamma\beta)_{\min}}{9.34\hat{\psi}_{\max}} \right).$$

This is the desired expression. □

**Lemma E.4.** *The covariances of the sufficient statistics, conditioned on $\Lambda$, are for all $i \neq j$,*

$$\mathbf{Cov}\left( \Lambda_i Y, \Lambda_j Y | \Lambda \right) = \Lambda_i \Lambda_j \mathrm{sech}^2(\psi^T \Lambda)$$

$$\mathbf{Cov}\left( \Lambda_i^2, \Lambda_j^2 | \Lambda \right) = 0.$$

*Proof.* The second result is obvious, so it suffices to prove only the first result. Clearly,

$$\mathbf{Cov}\left(\Lambda_i Y, \Lambda_j Y \middle| \Lambda\right) = \Lambda_i \Lambda_j \mathbf{Var}\left(Y|\Lambda\right) = \Lambda_i \Lambda_j \left(1 - \mathbf{E}\left[Y|\Lambda\right]^2\right).$$

Plugging into the distribution formula lets us conclude that

$$\mathbf{E}\left[Y|\Lambda\right] = \frac{\exp(\psi^T \Lambda + \phi^T \Lambda^2) - \exp(-\psi^T \Lambda + \phi^T \Lambda^2)}{\exp(\psi^T \Lambda + \phi^T \Lambda^2) + \exp(-\psi^T \Lambda + \phi^T \Lambda^2)} = \tanh^2(\psi^T \Lambda),$$

and so

$$\mathbf{Cov}\left(\Lambda_i Y, \Lambda_j Y \middle| \Lambda\right) = \Lambda_i \Lambda_j \left(1 - \tanh^2(\psi^T \Lambda)\right) = \Lambda_i \Lambda_j \operatorname{sech}^2(\psi^T \Lambda),$$

which is the desired result. $\qquad\square$

**Lemma E.5.** *If $\theta$ and $\theta^*$ are two feasible models, then for any $u$,*

$$\mathbf{E}_{\theta^*}\left[\mathbf{Var}_\theta\left(Y|\Lambda\right)\right] \le 3 \exp\left(-\frac{m\beta_{\min}^2 \gamma_{\min}^3}{8 \operatorname{artanh}(\gamma_{\max})}\right).$$

*Proof.* First, we note that, by the result of Lemma E.4,

$$\mathbf{Var}_\theta\left(Y|\Lambda\right) = \operatorname{sech}^2(\psi^T \Lambda).$$

Therefore,

$$\mathbf{E}_{\theta^*}\left[\mathbf{Var}_\theta\left(Y|\Lambda\right)\right] = \mathbf{E}_{\theta^*}\left[\operatorname{sech}^2(\psi^T \Lambda)\right].$$

Applying Lemma E.3, we can bound this with

$$\begin{aligned}
\mathbf{E}_{\theta^*}\left[\mathbf{Var}_\theta\left(u^T \Lambda Y|\Lambda\right)\right] &\le \left(\operatorname{sech}^2\left(\frac{m}{2}(\gamma\beta)_{\min}\gamma_{\min}^2\right) + \exp\left(-\frac{m(\gamma\beta)_{\min}\gamma_{\min}^2}{9.34 \operatorname{artanh}(\gamma_{\max})}\right)\right) \\
&\le \left(2 \exp\left(-\frac{m}{2}(\gamma\beta)_{\min}\gamma_{\min}^2\right) + \exp\left(-\frac{m(\gamma\beta)_{\min}\gamma_{\min}^2}{9.34 \operatorname{artanh}(\gamma_{\max})}\right)\right) \\
&\le 3 \exp\left(-\frac{m(\gamma\beta)_{\min}\gamma_{\min}^2}{9.34 \operatorname{artanh}(\gamma_{\max})}\right).
\end{aligned}$$

This is the desired expression. $\qquad\square$

# G   Additional Experimental Details

## G.1   Relation Extraction Experiments

### G.1.1   Systems

The original distantly-supervised experiments which we compare against as baselines–which we refer to as using the *if-then-return (ITR)* approach of distant or programmatic supervision–were implemented using DeepDive, an open-source system for building extraction systems.[5] For our primary experiments, we adapted these programs to the framework and approach described in this paper, directly utilizing distant supervision rules as labeling functions.

In the disease tagging user experiments, we used an early version of our new lightweight extraction framework based around data programming, formerly called DDLite [12], now Snorkel.[6] Snorkel is based around a Jupyter-notebook based interface, allowing users to iteratively develop labeling functions in Python for basic extraction tasks involving simple models. Details of the basic discriminative models used can be found in the Snorkel repository; in particular, Snorkel uses a simple logistic regression model with generic features defined in part over dependency paths[7], and a basic LSTM model implemented using the Theano library.[8] Snorkel is currently under continued development, and all versions are open-source.

### G.1.2 Applications

We consider three primary applications which involve the extraction of binary relation mentions of some specific type from unstructured text input data. At a high level, all three system pipelines consist of an initial *candidate extraction* phase which leverages some upstream model or suite of models to extract mentions of involved entities, and then considers each pair of such mentions that occurs within the same local neighborhood in a document as a *candidate relation mention* to be potentially extracted. In each case, the discriminative model that we are aiming to train–and that we evaluate in this paper–is a binary classifier over these candidate relation mentions, which will decide which ones to output as final true extractions. In all tasks, we preprocessed raw input text with Stanford CoreNLP[9], and then either used CoreNLP's NER module or our own entity-extraction models to extract entity mentions. Further details of the basic information extraction pipeline utilized can be seen in the tutorials of the systems used, and in the referenced papers below.

In the 2014 TAC-KBP Slot Filling task, which we also refer to as the News application, we train a set of extraction models for a variety of relation types from news articles [30]. In reported results in this paper, we average over scores from each relation type. We utilized CoreNLP's NER module for candidate extraction, and utilized CoreNLP outputs in developing the distant supervision rules / labeling functions for these tasks. We also considered a slightly simpler discriminative model than the one submitted in the 2014 competition, as reported in [2]: namely, we did not include any joint factors in our model in this paper.

In the Genomics application, our goal with our collaborators at Stanford Medicine was to extract mentions of genes that if mutated may cause certain phenotypes (symptoms) linked to Mendelian diseases, for use in a clinical diagnostic setting. The code for this project is online, although it remains partially under development and thus some material from our collaborators is private.[10]

In the Pharmacogenomics application, our goal was to extract interactions between genes for use in downstream pharmacogenomics research analyses; full results and system details are reported in [21].

In the Disease Tagging application, which we had our collaborators work on during a set of short hackathons as a user study, the goal was to tag mentions of human diseases in PubMed abstracts. We report results of this hackathon in [12], as well as in our Snorkel tutorial online.

### G.1.3 Labeling Functions

In general, we saw two broad types of labeling functions in both prior applications (when they were referred to as "distant supervision rules") and in our most recent user studies. The first type of labeling function leverages some weak supervision signal, such as an external knowledgebase (as in traditional distant supervision), very similar to the example illustrated in Fig. 1(a). All of the applications studied in this paper used some such labeling function or set of labeling functions.

The second type of labeling function uses simple heuristic patterns as positive or negative signals. For our text extraction examples, these heuristic patterns primarily consisted of regular expressions, also similar to the example pseudocode in Fig. 1(a). Further specific details of both types of labeling functions, as well as others used, can be seen in the linked code repositories and referenced papers.

### G.2 Synthetic Experiments

In Fig. 3(a-b), we ran synthetic experiments with labeling functions having constant coverage $\beta = 0.1$, and accuracy drawn from $\alpha \sim \text{Uniform}(\mu_\alpha - 0.25, \mu_\alpha + 0.25)$ where $\mu_\alpha = 0.75$ in the above plots. In both cases we used 1000 normally-drawn features having mean correlation with the true label class of 0.5.

In this case we compare data programming (DP-Pipelined) against two baselines. First, we compare against an *if-then-return* setup where the ordering is optimal (ITR-Oracle). Second, we compare against simple majority vote (MV).

In Fig. 3(c), we show an experiment where we add dependent labeling functions to a set of $m_{ind} = 50$ independent labeling functions, and either provided this dependency structure (LDM-Aware) or did

Figure 3: Comparisons of data programming to two oracle methods on synthetic data.

not (Independent). In this case, the independent labeling functions had the same configurations as in (a-b), and the dependent labeling functions corresponded to "fixes" or "reinforces"-type dependent labeling functions.