[Reviews · NeurIPS 2016]

Reviewer 1

Summary

The authors present a principled mechanism to create large datasets through series of fuzzy labeling functions. They propose to model the noise inherent in these fuzzy labeling functions to denoise the dataset. This technique may help non-experts to quickly generate datasets for machine learning purposes. The concept is very interesting and promising.

Qualitative Assessment

I thought this is very interesting and particularly a problem in my domain of interest, healthcare. In the clinical domain, much knowledge is in the minds of clinical experts and labeling is very expensive and not scalable. In addition, there is vast uncertaintyin the quality of the captured data. For example, there may be a structured field called "smoking" status in a health record. One may think that whatever is input into this field is reliable. However the reality is that the workflow, location in the electronic health record, and other issues may make the field noisy and less reliable. This general technique could help to formalize the noise process. The general technique is promising and something that I would consider using to help extract domain knowledge from my physician experts toward labeling documents. MAJOR - add some documentation on the size of the datasets used in Table 2. I am guessing that they are large enough to use these labeling functions to generate the claimed "large training sets." MINOR - improve description of table 2. add definition of LF (labeling function). - improve description of table 1. define DP (data programming) and what the bold numbers represent. - on lines 215-218. the authors explain fixing and reinforcing. Add textual description of similar and exclusive (though they are somewhat obvious). - add a sentence after line 137 enumerating what -1, 0, and 1 mean. The authors do this later but I found that I had to flip around the paper to get the definitions as I was reading it. - confusing on line 135 to say that labeling function provides non-zero label. In implementation as in figure 1, it does provide zero values.

Confidence in this Review

2-Confident (read it all; understood it all reasonably well)


Reviewer 2

Summary

The authors propose a technique they call "Data Programming", which entails algorithmic generation of (noisy) training data. This extends the paradigm of “distant supervision” by requiring the specification of generalized labeling functions. I think this is an interesting paper that describes a relatively novel paradigm. Furthermore, the empirical results are convincing. I particularly appreciated the initial experience with real world ‘users’ of the paradigm (although these are indeed quite preliminary and not much is reported regarding this). I can see this approach/abstraction being widely useful.

Qualitative Assessment

Specific comments: * Seems like the \alpha_i’s should be class-dependent (so sens./spec. rather than just overall accuracy). In particular, it would seem intuitively rather easy to write high-recall/low-specificity labeling functions, but this granularity would be lost on the model as currently specified. * The assumption stated in (5) seems rather strong and I would have appreciated a bit more discussion around the implications. In particular, does this imply that the labeling functions are unbiasedly reliable across all inputs? In practice this seems very unlikely to me. In turn, this brings into question the practical applicability of the derived bounds. * I did not understand this claim “Conventional wisdom states that deep learning methods such as RNNs are prone to overfitting, thus rendering them ineffective over distantly-supervised training sets”. I *think* the authors are saying that noisy data leads to overfitting? But actually, assuming random noise anyway (a big assumption, admittedly), noisy labels would probably lead to *less* overfitting. Indeed, DS applied to very large datasets actually seems to be an intuitively good fit for training deep nets. * W.r.t. the user experiment, it seems slightly misleading to emphasize that “no feature engineering was performed”, when in some sense the labeling functions (manually designed) *are* features — just very good ones, no? I’m also curious what would have happened if you had instead allowed the researchers to spend those 8 hours instead just labeling training data. I wouldn’t be shocked if this resulted in similar improvements (that’s quite a bit of time!). Small comments —— * I think the notation would be more explicit with instance indices throughout (e.g., x_j in eq 3, rather than just x). * In Thm. 1 the authors assume m=2000 labeling functions. This seems like a lot, no?

Confidence in this Review

2-Confident (read it all; understood it all reasonably well)


Reviewer 3

Summary

This paper describes an approach to creating large-scale data sets by providing with a set of manually created rules. Data creators write a set of labeling functions, rather than annotating all the large-scale data sets. Then, regarding the labeling functions as a description of noisy generative models, the approach recovers the parameters of the model to “denoise” the data set. Experimental results show that the approach was effective over several data sets.

Qualitative Assessment

- The title says that the aim of this paper is to create large training sets quickly. How large data sets are created and how fast? - You should evaluate the quality of the created data sets. For example, you can compare the relation extracted data set created by Data Programming and the 2014 TAC-KBP Slot Filling training data set based on the same news articles and the annotation guideline. - It is very much unclear how the labeling functions are created. How many people created how many rules with referring to which part of training data? Did they see the correct labels in the data set to created labeling functions for the TAC-KBP data? - Some theoretical results, e.g. Theorem 1, seem to be related to PAC learning. You should refer to the PAC learning theory. - This study is also closely related to Active Learning. You should prove the advantage to the Active Learning approach.

Confidence in this Review

2-Confident (read it all; understood it all reasonably well)


Reviewer 4

Summary

The authors describe a system for labeling datasets using expert-provided labeling functions, with the idea that labels can be generated more quickly this way than by requiring labels for individual instances. Two joint distributions (over label-function outputs and true-labels) are proposed, one that assumes conditional independence of the label-functions given the true label and one that factors in dependencies between label-functions. Theorems are provided showing expected error bounds when learning the parameters of these distributions and the parameters of a linear classifier using SGD. Table 1 reports experimental results. Their ITR method uses several labeling-functions to produce a single "best" label. The data-programming (DP) method uses all of these labeling-functions as described above. DP-based models perform better than ITR-based models on three datasets.

Qualitative Assessment

Reducing the cost of labeling datasets is obviously a worthy goal. This paper is a nice attempt at doing that. This is more about future work, but I wonder how well it could be extended to non-NLP tasks. Writing short, simple programs to label images with high accuracy, for example, seems much more challenging. I wonder how well the method will work when the experts are not able to provide labeling-functions that have 30-50% coverage and 80-90% accuracy (see lines 160-168). A few minor questions/points for improvement: - Is P(Lambda=lambda(x)) in equation 2 a marginal? You're summing out Y, right? - What is pi^* in Eqs. 4 & 5 and in other places? - You didn't define M in line 223 (and it shows up other places). - Section 1 has lots of mentions of unspecified "previous/prior approaches". It would be nice to have some names of specific approaches before getting to Section 2. - Also in Section 1 are the phrases "generating training sets" and "generative model for a training set". That's somewhat confusing since it implies full training sets (features and labels) are being generated, not just labels.

Confidence in this Review

2-Confident (read it all; understood it all reasonably well)


Reviewer 5

Summary

The paper talks about a fundamental paradigm shift in building supervised ML systems. Generating ground truth (labeled dataset) to be used as training sets for different ML algorithms and frameworks through crowdsourcing or otherwise is an expensive step since it may not always be available. The data programming paradigm discussed in this paper instead focuses on how to improve existing ML systems by using unlabeled data along with some broad labeling heuristics given by domain experts (which are albeit noisy). Experimental results indicate that data programming can significantly improve supervised ML systems under certain conditions.

Qualitative Assessment

This is a interesting work. The author motivates the problem regarding how availability of large labeled training sets may be hindrance to several supervised ML systems and deep learning techniques. And data programming can be an interesting approach here. Also the user study indicating how researchers find it easy to write labeling heuristics instead of generating ground truth through crowdsourcing or otherwise is a good indication of the utility of this technique. The writing is clear and easy to follow, the experiments are thorough. Following are a few pointers/suggestions which I think the authors should follow up on: 1. There was an older NIPS paper by Rich Caruana regarding “Do deep nets really need to be deep” which explores the ideas of model compression. Here, a very small labeled training data is used to build a large model which then produces massive amounts of synthetically labeled dataset from massive unlabeled datasets to train compressed models. I think this should also be cited in the related work as it has been successfully deployed in many practical systems. This brings us to two significant advantages of data programming: In the former work it was mentioned that the synthetically labeled dataset needs to be around 10 times larger than the actual labeled ground truth. There are no such restrictions with respect to data programming. Second, data programming seems to improve performance of supervised ML algorithms without requiring any additional training instances which is counter-intuitive. 2. The results indicate that the coverage of the labeling functions can be as low as 7% which is quite surprising? Is their any further insight into this? 3. My one concern is that the labeling heuristics might need to have very high accuracy (as given by alpha values) which may not always be possible. It will be interesting to see what happens when labeling functions have poor to moderate accuracy even when they are independent? Nevertheless this work provides an interesting direction.

Confidence in this Review

2-Confident (read it all; understood it all reasonably well)